# Quantifying hydrologic connectivity of wetlands to surface water systems

Ali A. Ameli[1], Irena F. Creed[1]

[1]Department of Biology, Western University, London, Ontario, Canada

*Correspondence to*: Irena Creed (icreed@uwo.ca)

**Abstract.** Hydrologic connectivity among wetlands is poorly characterized and understood. Our inability to quantify this connectivity compromises our understanding of the potential impacts of wetland loss on watershed structure, function and water supplies. We develop a computationally efficient physically-based subsurface-surface hydrologic model to characterize both the subsurface and surface hydrologic connectivity of "geographically isolated" wetlands and explore the time and length variations in these connections to a river within the Prairie Pothole Region of North America. Despite a high density of geographically isolated wetlands (i.e., wetlands without surface inlets or outlets), modeled connections show that these wetlands are not hydrologically isolated. Subsurface connectivity differs significantly from surface connectivity in terms of timing and length of connections. Slow subsurface connections between wetlands and the downstream river originate from wetlands throughout the watershed, whereas fast surface connections were limited to large events and originate from wetlands located near the river. This modeling approach provides first ever insight on the nature of geographically isolated wetland subsurface and surface hydrologic connections to rivers, and provides valuable information to support watershed-scale decision making for water resource management.

**Keywords:** wetland, geographically isolated wetlands, hydrologic connectivity, surface water, groundwater, Prairie Pothole Region

## 1 Introduction

Enhanced protection of wetlands is urgently needed (Dixon et al., 2016). Geographically isolated wetlands (GIWs) (Tiner, 2003) are in particular need of protection, as these are small features vulnerable to filling or drainage (Cohen et al., 2016). Geographical isolation does not imply hydrologic, biogeochemical or biological isolation (Mushet et al., 2015;Leibowitz, 2015;Marton et al., 2015;Rains et al., 2015). Rather, GIW hydrologic connections vary in time and space, and these connections can occur through persistent but slow velocity groundwater pathways (McLaughlin and Cohen, 2013;Brannen et al., 2015;Hayashi et al., 2016) or transient but fast surface water pathways via mechanisms analogous to fill-and-spill runoff generation (Rains et al., 2006;Shaw et al., 2012;Leibowitz and Vining, 2003). Wetland ecosystem functions (e.g., collecting, storing, filtering or discharging water, sediments and solutes) arise from the cumulative effects of these diverse hydrologic connections (Cohen et al., 2016). For example, a lack of persistent surface connection from wetlands to rivers leads to the restriction of material and organism exchanges in landscapes; this restriction allows GIWs to influence downslope resources (US-EPA, 2015) by enhancing flood regulation and surface water quality (Golden et al., 2014;Leibowitz, 2003). Subsurface connections of wetlands, on the other hand, also affect surface water resources (Cook and Hauer, 2007;Euliss et al., 2004). For example, groundwater connections between GIWs and rivers can regulate the groundwater table, stabilize base flows and change base flow chemistry (McLaughlin et al., 2014). Characterization of both surface and subsurface connections are crucial for the provision of important aquatic ecosystem services (Winter and LaBaugh, 2003).

The quantification of wetland connectivity (i.e., subsurface and surface connections among wetlands and between wetlands and rivers) remains a significant scientific challenge (US-EPA, 2015;Cohen et al., 2016;Golden et al., 2017). Our inability to quantify wetland connections compromises our understanding of (1) the role of the continuum in the timing and length of wetland connections on landscape functions; (2) the effects of environmental stressors (e.g., climate and land use or land cover change) on this continuum of wetland connections; and (3) the effects of human alterations of wetland connections on downstream waters. More importantly, our inability to quantify wetland connectivity may lead to inappropriate management decisions regarding wetland protection *vs.* removal (Van Meter and Basu, 2015). Indeed, decisions to protect or drain GIWs are often made based on their proximity to a major drainage network (Cohen et al., 2016). The quantification of wetland connectivity is required to enable prioritization of wetland protection and restoration, where the optimum location of drainage or restoration of wetlands and the hydrologic, biogeochemical and biological functions of each individual drained or restored wetland can be evaluated (Golden et al., 2017).

Effective quantification of wetland connectivity requires a modeling tool that can explicitly take into account various types (surface and subsurface) and lengths of wetland connectivity under different climate and land use or land cover scenarios. Process-based modeling tools (e.g., SWAT) are useful for assessing *aggregated* impacts of wetland connectivity on watershed-scale targets (e.g., watershed-scale phosphorus load or peak flow reduction) (Shrestha et al., 2012). However, these modeling tools cannot explicitly consider individual wetlands and characterize their links to other wetlands and rivers, particularly in wetland dominated systems (Golden et al., 2014); these considerations are necessary if one intends to prioritize protection and restoration of individual wetlands (Golden et al., 2017). In contrast, numerical physically-based groundwater-surface water flow and transport modeling tools have the ability to sufficiently incorporate subsurface and surface wetland connections (Golden et al., 2014). These models, however, are typically grid-based (i.e., discrete) and require high level modeling expertise and high computational power, particularly for the simulation of watershed-scale subsurface connections. This is particularly true when these models are confronted with a range of wetland sizes; although a small sub-watershed system can be discretized so that a single wetland falls within a single grid cell, incorporation of wetlands with a variety of sizes in larger wetland-dominated watersheds is challenging and prone to discretization artifacts (Golden et al., 2014). Furthermore, watershed-scale tracking of water that flows from a wetland to other local or regional surface waters within such grid-based modeling systems can be challenging and inefficient, particularly when the size of grid spacing increases to reduce computational cost (c.f., Salamon et al. (2006)).

There is currently no physically-based model that adequately captures watershed-scale wetland connectivity. Recently, Ameli and Craig (2014) developed a grid-free physically-based integrated flow and transport scheme for simulation of 3D groundwater-surface water interactions. This 3D grid-free model is scale-independent, implying that it has the potential to efficiently simulate "watershed-scale" (or even larger scale) groundwater-surface water interaction and subsurface connections among individual wetlands and between each wetland and regional surface waters without domain discretization artifacts. Here, we use this model to map watershed-scale *subsurface* connections, and then link this model with a physically-based transient surface flow routing simulator to map watershed-scale *surface* connections.

Specifically, for a large watershed dominated with GIWs within the prairie pothole region (PPR) of North America, we: (1) Assess the performance of the 3D groundwater-surface water interaction model at regional scales against ground-based (hydrometric, tracer, isotopic) measurements; (2) Compare the distribution of time and length characteristics of simulated wetland subsurface vs. surface connections; (3) Explore if the shortest distance of a wetland to other surface water bodies is an appropriate indicator of wetland connectivity; and (4) Explore if our findings can be extended to the other parts of PPR. Our

wetland connectivity modeling approach fills a fundamental gap toward advancing the science and management of wetland hydrologic, biogeochemical and biological connectivity in landscapes.

## 2 Material and methods

### 2.1 Experimental watershed

The Beaverhill watershed comprises 4405 km$^2$ and is situated on the north-western edge of the Prairie Pothole Region (PPR) (Figure 1). The watershed is centered on the Cooking Lake moraine and drains into the North Saskatchewan River near Edmonton, Alberta, Canada. The watershed is dominated by natural forest within the moraine and agriculture (predominately grassland and pastureland) outside the moraine, with a considerable amount of urban and industrial development near the North Saskatchewan River and the city of Edmonton (Serran and Creed, 2015). The Cooking Lake moraine was recently recognized as a biosphere reserve by UNESCO and contains Beaverhill Lake located in the eastern portion of the watershed, which is recognized as a wetland of international importance by the RAMSAR convention on wetlands.

The climate, geology and topography have collectively created a hydrologic system dominated by numerous lakes and wetlands as well as only a few intermittent or slow-moving streams. The climate is continental with cold winters and warm summers. Based on the 40-year (1974-2014) climatic data collected at the Edmonton International Airport, the average January temperature is -13.5 °C and the average July temperature is 15.9 °C (http://climate.weather.gc.ca/). Average annual precipitation is 483 mm, of which almost 70% falls as rain between May and September, a period when the potential evapotranspiration is as large as 450 mm. This means that there is generally little surface runoff, although spring snowmelt can be an important contributor of local runoff to a wetland.

The geology is dominated by glacial deposits resulting from the Pleistocene continental glaciers. Three till sequences with variable thickness were left as a result of the last glaciation. The higher permeability shallow till often extends from the land surface down to below the average position of the water table. For example, within the St. Denis National Wildlife Area located 500 km east of the Beaverhill watershed, van der Kamp and Hayashi (2009) reported a thickness of 4-5 m with an approximate range of saturated hydraulic conductivity between $2 \times 10^{-2}$ to 2 m/d, and a saturated hydraulic conductivity of less than $2 \times 10^{-3}$ m/d below this layer. While the transmission rate of water from the shallow to deeper geological deposits is slow, the moraine still serves as an important source of groundwater recharge in the area, with the annual groundwater recharge rate within the Beaverhill moraine estimated to vary spatially from $5 \times 10^{-5}$ m/d to $1.9 \times 10^{-4}$ m/d (Barker et al., 2011;Sass et al., 2014). The surficial bedrock geology is predominantly characterized by the Horseshoe Canyon Formation that is composed of fine to medium grained sediments, inter-fingered within muddy, transgressive sediments (Barker et al., 2011). The lithology of the surficial bedrock suggests that Ca and Mg are dominant weathering-derived products at the Beaverhill watershed.

The topographic relief ranges from a high of 812 m a.s.l. in the moraine to a low of 586 m a.s.l. along the North Saskatchewan River, and reflects glacial depositional processes comprising knob, kettle and hummocky formations in the moraine surrounded by flat to rolling areas in lower elevations.

### 2.2 Mapping wetlands

A total of 130,157 wetlands were delineated based on the assumption that there is a strong association between terrain depressions and wetland occurrence. A Light Detection and Ranging (LiDAR) DEM captured in 2009 with a horizontal resolution of 3 m and an estimated vertical accuracy of 15 cm was used to map the probability of depression using the approach offered by Lindsay and Creed (2005). The depression probability map was then segmented into image objects using the multi-

resolution segmentation algorithm (Baatz and Schäpe, 2000). Average depression probability values were used to classify objects
as wetland or non-wetland, and adjacent classified wetland objects were dissolved into a single wetland (Serran and Creed,
2015). Note that this method delineated both potential wetlands without surface outlets (i.e., GIWs) and with surface outlets;
given the sparsity of permanent streams in the watershed most of the delineated wetlands can be considered *a priori* as GIWs.
This is consistent with the observation of GIW predominance in the PPR (Cohen et al., 2016).
**2.3 Modelling wetland connectivity**

7        A 3D steady-state groundwater-surface water interaction model was used to simulate watershed-scale subsurface flow

and velocity fields as well as to calibrate infiltration rate (Sect. 2.3.1). The mathematical formulation, solution parameters and
boundary conditions used in the model as well as the incorporation of the map of wetlands within the model algorithm are
explained in Appendix A. The calibrated infiltration rate was combined with meteorological data in a 2D transient overland flow
model to simulate surface flow routing (via a fill and spill mechanism) and to simulate watershed-scale surface water level and
velocity fields (Sect. 2.3.2). This one-way linked subsurface-surface model does not consider thoroughly the potential feedbacks
from subsurface flow on surface flow routing. Finally, continuous watershed-scale maps of subsurface and surface velocity were
generated and coupled with the wetland map to track water movement and to determine if water issued from an individual
wetland reached a discharge surface water body (e.g., North Saskatchewan River) via subsurface or surface pathways (Sect.
2.3.3). Note that to characterize the map of connectivity to North Saskatchewan River, the river boundary includes a 500 m
buffer around the original line segment of the river which was obtained from a standard hydrography dataset.
**2.3.1 Groundwater-Surface water interaction model**

19        The 3D groundwater-surface water interaction model used a free boundary condition to determine the location of the

water table (Bear, 1972;Bresciani et al., 2016). This condition was iteratively imposed using a recharge-water table depth
relation scheme (Ameli and Craig 2014) that creates a spatially variable recharge rate and enables delineation of discharge areas
where the water table reaches the land surface. This scheme assumed a steady-state subsurface flow and a hydraulic connection
between groundwater and wetland water levels. The assumption of steady-state subsurface flow is strongly supported by
empirical groundwater table observations collected from the closest piezometer at the Vegreville Environment Center station
(located 15 km east of the Beaverhill watershed boundary), where the water table varied with a coefficient of variation of < 0.9%
in 2009 (a year when observations were used to develop the steady-state groundwater-surface water interaction model), and a
coefficient of variation of 4% over 32 years (August 1985-July 2016) (Figure 2a). In another piezometer at the Barrhead
Environment Center station located 65 km west of the Beaverhill watershed boundary, water table varied even less with a
coefficient of variation of ~0% during 2009, and a coefficient of variation of 0.01% over 40 years (1977-2016) (Figure 2b). The
assumption of hydraulic connection between groundwater and wetland water levels is supported by previously reported empirical
observations at the St. Denis National Wildlife Area, 500 km east of the Beaverhill watershed, that showed a maximum
difference of less than 10 cm between the groundwater level at a piezometer located 7 m from the wetland edge and the wetland
water level (van der Kamp and Hayashi, 2009).

34        The model was calibrated using saturated hydraulic conductivities of the two-layer unconfined aquifer and actual

infiltration rate (see Appendix A for more details). The model performance was assessed for its ability to map groundwater
discharge vs. recharge areas and subsurface connections using multiple lines of corroborating hydrometric, chemical and isotopic
evidence. First, we compared the simulated groundwater discharge and recharge areas to one derived from hydrometric
measurements. We used measurements of hydraulic heads in 1,413 artesian groundwater wells installed in the bedrock and
screened 30 to 80 m below the land surface (http://aep.alberta.ca/water/reports-data/alberta-water-well-information-database/)
and used a kriging approach to map the potentiometric surface throughout the entire watershed for summer 2009 (see Figure 1
for the location of the artesian wells). Groundwater discharge and recharge areas are inferred as areas wherein potentiometric
surface is above and below land surface, respectively (Barker et al. (2011).
Second, we determined if the simulated groundwater discharge and recharge areas had different chemical signatures. It
is known that the concentration of weathering-derived products (such as Ca and Mg), chemical measures affected by weathering
processes (such as alkalinity and electric conductivity, EC) and total dissolved solids (TDS) can be enhanced along the
subsurface flow pathways as transit time, and therefore contact time with rock, increases (e.g., Burns et al., 2003;Maher and
Druhan, 2014;Godsey et al., 2009;Cook and Hauer, 2007;Ameli et al., 2017). This implies that the concentration of Ca, Mg,
alkalinity, EC and TDS in groundwater wells and surface water bodies (e.g., wetlands, lakes) located within discharge areas will
be higher than recharge areas (Cook and Hauer, 2007;Euliss et al., 2004;Barker et al., 2011). We mapped Ca, Mg, alkalinity, EC
and TDS measurements of 121 shallow (< 10 m deep) groundwater wells located throughout the watershed provided by Alberta
Water Well Information Database (Figure 1), and for 208 surface waters including lakes and wetlands located throughout the
watershed (Figure 1) in summer 2009. For the latter, water samples were collected at a depth of 1 meter using an integrated
sampling tube at the center of small, shallow wetlands and 100 meters from the shores of large, deep lakes. A non-parametric
Wilcoxon rank sum test was used to see if Ca, Mg, alkalinity, EC and TDS concentrations or values were significantly different
(higher) at groundwater wells and wetlands located in discharge areas compared to recharge areas based on comparing the $p$
values of the statistical tests to the significance level of 0.10. At any $p$ values larger (smaller) than 0.10 we accept that the
concentrations at discharge and recharge areas are (are not) from distributions with equal medians.
Third, we determined if the simulated discharge and recharge surface water bodies had different isotopic signatures. It is
known that $^{18}O$ and $^2H$ signatures vary between discharge and recharge areas; indeed, discharge waters that have more old water
will have different isotopic signatures than recharge waters that have new water only, either from direct precipitation or indirect
precipitation via overland flow (McGuire and McDonnell, 2006;Kirchner, 2016;McDonnell and Beven, 2014). This implies that
the average isotopic concentration (ratio ‰) in discharge wetlands shows greater deviation from the average watershed
concentration of surface waters than the deviation of recharge wetlands. We mapped isotopic $^{18}O$ and $^2H$ signatures in samples
collected for the same 208 surface waters for which the chemical tracers were sampled (Figure 1) in summer 2009, and used
Wilcoxon rank sum test to assess the potential differences in isotopic signatures between discharge and recharge wetlands.
The first line of corroborating evidence (hydrometric) was used to calibrate the groundwater-surface water interaction
model, and the second and third lines of corroborating evidence (chemical and isotopic) were used to evaluate the performance
of the model.
**2.3.2 Surface "Fill and Spill" overland flow model**
The 2D transient fill and spill surface flow routing approach within the numerical physically-based HydroGeoSphere
model (Therrien et al., 2008) was used to simulate watershed-scale surface water level and overland flow routing and ultimately
to determine the surface connectivity of wetlands using a transient water particle tracking scheme. The 2D surface of the
watershed was discretized into 22,383 grid points (43,836 triangular elements). The parameters regarding time-discretization
were: maximum time step = 8640 sec; initial time step = 1800 sec; maximum time step multiplier = 1.5; and minimum time step
multiplier = 0.5. A critical depth boundary condition was assigned to the grid points representing the location of the Beaverhill
Creek monitoring station (Figure 1) where stream flow observations were available. A no-flow boundary condition was assigned
to the watershed boundaries.
The 2D overland flow model was calibrated using the Manning roughness coefficients (Manning et al., 1890) in $x$ and $y$
directions ($n_x$ and $n_y$) as well as rill depth. The former is an empirically derived coefficient, which is dependent on surface
roughness and surface cover, and the latter represents the depth that must be filled at each point before any lateral surface flow
can occur. Frei and Fleckenstein (2014) suggested that the implementation of an acceptable uniform value for rill depth within
the HydroGeoSphere overland flow simulator leads to an accurate prediction of watershed-scale surface flow routing. Stream
flow measurements were available from 1975 to 1986 at the Beaverhill Creek monitoring station (Figure 1). The largest runoff
events when the stream flow measurements were available occurred from April 1 to August 1, 1983, and the smallest runoff
events occurred from April 1 to August 1, 1985. Model calibration was conducted to match observed and simulated stream flow
during the wettest period (April 1 to August 1 1983). Model validation was then conducted to match observed and simulated
stream flow during the driest period (April 1 to August 1 1985).  The inputs to the models included daily precipitation,
evapotranspiration (after Morton 1978, Morton 1983) and snow water equivalent (after Sturm et al. 2010) from data collected at
the Vegreville Environment Center meteorological station (15 km east of the Beaverhill watershed boundary) as well as the
steady-state infiltration rate obtained using the 3D groundwater-surface water interaction model. We did not have access to
evapotranspiration data before 2000 (including the calibration and validation periods). Instead, we used the calculated
evapotranspiration time history of 2015 for the same period (April 1 to August 1, 2015), because the monthly average humidity,
maximum air temperature and minimum air temperature were almost similar in 1983, 1985 and 2015.
The calibrated 2D overland flow model was then used to simulate surface flow routing from April 1 to August 1, 2009
as well as from April 1 to August 1, 2013 using meteorological data for these periods as inputs to the model. The chosen time
periods include the smallest and the largest cumulative net water (precipitation-minus evapotranspiration) depth observed at the
Vegreville station since 2000, reflecting the minimum and maximum probability of occurrence of surface flow and therefore
surface connections among wetlands since 2000. Once the surface flow routing model was solved, the discretized surface water
velocities in x and y directions at each grid point and each time step were extracted. Continuous maps of surface water velocity
in x and y directions throughout the watershed were then approximated by interpolating the discretized surface water velocities.
A Fourier-based interpolation scheme with 10,000 Fourier series terms was used to complete the interpolation process and
generate the continuous maps of surface velocity in x ($V_s^x(x, y, t)$) and y ($V_s^y(x, y, t)$) directions for the entire watershed; the
overall correlation coefficient between estimated velocities using the interpolation method and original modeled velocities at
each grid point and time step was $r^2 = 89\%$ (p < 0.001).
**2.3.3 Subsurface and surface wetland connections**
The continuous maps of subsurface steady-state ($V_m^x(x, y, z)$, $V_m^y(x, y, z)$, $V_m^z(x, y, z)$, Eq. A.4) and transient surface
velocity ($V_s^x(x, y, t)$, ($V_s^y(x, y, t)$) were used to track water particles and generate a connectivity map using water particle
tracking approach as follows.
In the subsurface domain: $x_p(t) = x_p(t - \Delta t) + V_m^x(x_p(t - \Delta t), y_p(t - \Delta t), z_p(t - \Delta t)) * \Delta t$     (1a)
$y_p(t) = y_p(t - \Delta t) + V_m^y(x_p(t - \Delta t), y_p(t - \Delta t), z_p(t - \Delta t)) * \Delta t$
$z_p(t) = z_p(t - \Delta t) + V_m^z(x_p(t - \Delta t), y_p(t - \Delta t), z_p(t - \Delta t)) * \Delta t$

In the surface domain:     $x_p(t) = x_p(t - \Delta t) + V_s^x(x_p(t - \Delta t), y_p(t - \Delta t), t - \Delta t) * \Delta t$     (1b)
$y_p(t) = y_p(t - \Delta t) + V_s^y(x_p(t - \Delta t), y_p(t - \Delta t), t - \Delta t) * \Delta t$

where $x_p(t)$, $y_p(t)$ and $z_p(t)$ are the position of the particle $p$ at time $t$, and $x_p(t - \Delta t)$, $y_p(t - \Delta t)$ and $z_p(t - \Delta t)$ are the
position of the particle $p$ at time $t - \Delta t$. In the particle tracking algorithm, a small value of time step ($\Delta t = 0.1$ day) was
assumed to ensure precise calculation of particle location and movement. A similar particle tracking method was recently used in
Ameli et al. (2016b) and Ameli et al. (2016a).

5        To simulate hydrologic connection among wetlands or between wetlands and the North Saskatchewan River, water

particle release points in the tracking approach were placed uniformly with 500 m spacing along the land surface. This placement
meant that there was a possibility that not all wetland connections were captured (i.e., water particle release points and small
wetlands in between the 500 m placement would have been missed); nonetheless, it allowed for a consistent comparison of the
general trend in subsurface and surface connections of wetlands. The generated connections were used to estimate subsurface
and surface transit times ($\tau$) and flowpath lengths ($l$) that were then fitted with a Gamma distribution (which provided the best fit,
data not shown) to generate the transit time distribution ($\rho_\tau$) and flowpath length distribution ($\rho_l$):

12          $\rho_\tau(\tau) = \dfrac{a(\frac{\tau}{\tau_0})^{a-1}}{\tau_0 \Gamma(a)} e^{-a(\frac{\tau}{\tau_0})}$    &    $\rho_l(l) = \dfrac{a(\frac{l}{l_0})^{a-1}}{l_0 \Gamma(a)} e^{-a(\frac{l}{l_0})}$                       (2)

where $\tau_0$ and $l_0$ are the mean transit time and length of hydrologic connection among wetlands or between wetlands and the
North Saskatchewan River, $\Gamma(a)$ is the Gamma function and $a$ is the Gamma shape parameter. The time (and length)
characteristics of the simulated subsurface and surface connections of wetlands were then compared.

16        We calculated subsurface and surface transit times and flow contributions of each wetland to the North Saskatchewan

River. The relationship between distance to the river and transit times or flow contributions of wetlands can determine if distance
is an effective proxy for hydrologic connectivity to the river.

19        The simulated subsurface connections were tested by correlating the simulated transit time of water particles discharged

into discharge wetlands and the concentration of chemical tracers (Ca, Mg, EC and TDS) in the discharge wetlands. We
determined if the discharge wetland which received particles with longer subsurface transit times had higher concentrations or
values of Ca, Mg, alkalinity, EC and TDS. The pathlines (and their corresponding transit times) of water particles discharged
into each discharge wetland, were calculated by back tracking from 100 uniformly-distributed particle release points located at
each discharge wetland. The transit time corresponded to each discharge wetland was then calculated as the average of transit
times of all 100 particles.

26        The generalizability of our findings to the entire PPR was assessed by comparing climate, geology and topography of

the Beaverhill watershed to the other parts of the prairie potholes regions of North America. We compared monthly-averaged
climatic measures (including precipitation, evapotranspiration, snow water equivalent and temperature) of the Beaverhill
watershed to the entire Prairie Pothole Region. We compared the average porosity and permeability of the Beaverhill watershed
to the average of these measures in the entire Prairie Pothole region. We also compared the distribution of observed shortest
distances among nearest wetland neighbour and distances between wetlands and stream network at the Beaverhill watershed to a
prairie pothole landscape in North Dakota. We obtained wetland polygons in North Dakota from the National Wetlands
Inventory (https://www.fws.gov/wetlands/ and stream polylines from the National Hydrography Dataset: http://nhd.usgs.gov/).
To do the latter comparison, we used Quantile-Quantile plot as a graphical non-parametric method for comparing probability
distributions of two unpaired samples by plotting their quantiles against each other. If the two distributions being compared are
statistically similar, the theoretical line of the Quantile-Quantile plot will be $y = x$ and the quantile pairs will approximately lie
on this line.  If the distributions are not statistically similar, the theoretical line of the Quantile-Quantile plot will be a linear line,
but not following $y = x$.
**3. Results**

## 3.1 Groundwater-Surface water interaction model

The calibrated values of the saturated hydraulic conductivities are $10^{-1}$ m/d and $10^{-3}$ m/d for the top and bottom layers respectively. The calibrated value for infiltration rate (which here is equal to maximum groundwater recharge rate) is $1 \times 10^{-4}$ m/d.

The simulated groundwater discharge/recharge map is consistent with the map of groundwater discharge/recharge inferred from measured hydraulic head (potentiometric surface) in piezometric wells (Figure 3). Figure 3a shows the spatial distribution of groundwater discharge (negative) and recharge (positive) fluxes along the land surface obtained using the groundwater-surface water interaction model. Figure 3b shows the distance of potentiometric surface from the land surface, with negative values (above land surface) representing discharge areas and positive values (below land surface) representing recharge areas, with the larger negative (or positive) values equal to larger discharge (or recharge) potential. The correlation coefficient between simulated groundwater fluxes at the land surface and the distance of potentiometric surface above and below land surface is 75% (p < 0.001). Figure 3 also shows the predominance of recharge area within the moraine and discharge area outside of the moraine.

The performance of the model was also assessed using chemical and isotopic tracer data. The concentrations or values of chemical tracers (Ca, Mg, EC and TDS) of water in shallow groundwater wells are different between the simulated discharge and recharge areas (p < 0.10) (Table 1a). The average concentrations of chemical tracers are higher in the simulated discharge areas than the simulated recharge areas. In addition, the concentrations or values of all chemical tracers (except Mg) in simulated discharge wetlands are statistically different from simulated recharge wetlands (p < 0.10) (Table 1b). The average concentrations of all chemical tracers are higher in the simulated discharge wetlands than the simulated recharge wetlands. Higher concentrations of weathering products reflect the existence of longer pathlines with larger transit times within simulated discharge areas.

The average concentration of isotopic tracers ($^{18}O$ and $^{2}H$) in simulated discharge and recharge wetlands are significantly different ($p < 0.001$) (Table 2). Also for $^{18}O$ ($^{2}H$), the average isotopic concentration in simulated discharge areas deviates 3‰ (8‰), whereas the average isotopic concentration in simulated recharge areas deviates only 0.9‰ (3‰) compared to the watershed average isotopic concentrations. This reflects a mixture of old and new waters in simulated discharge wetlands, but mostly new waters in simulated recharge wetlands, which is consistent with our expectations.

## 3.2 Surface "Fill and Spill" overland flow model

The manually calibrated values of the uniformly-distributed Manning roughness coefficient is $0.05 \frac{sec}{m^{1/3}}$ (equal in both $x$ and $y$ directions) and the rill storage height is 0.001 m. Figure 4a shows that during the calibration period the simulated vs. observed stream flow at the Beaverhill Creek near the mouth measurement station for the major summer rainfall period from April 1 to August 1 1983 are significantly correlated ($r^2 = 0.87$, $p < 0.001$). Two statistical tests, including Wilcoxon Rank Sum (equality of median) and Levene (equality of variance), also suggest that the median and variance of both simulated and observed hydrographs are similar ($p$ values are 0.44 and 0.95, respectively. During the validation period, the simulated vs. observed stream flow at the mouth of Beaverhill Creek for the driest period (from April 1 to August 1 1985) were also correlated ($r^2 = 0.45$, $p < 0.001$) (Figure 4b). We did not expect that the surface flow model would exactly simulate the hydrograph in 1983 and 1985, as we used evapotranspiration data from 2015 for these two years (as explained in section 2.3.2). This simplification could have affected the simulated hydrograph shape leading to an earlier second peak in the hydrograph of 1983 (Figure 4a). We think this simplification would have had minimal effect on the simulated connectivity lines, as at the end of the simulation period of 1983, the cumulative simulated flow ($2.4 \ 10^7$ m$^3$) at the measurement station was only 7% less than the cumulative observed flow (2.6

$10^7$ m$^3$). Indeed, for the particle tracking scheme used to characterize the surface connectivity map, it did not make a substantial
difference if the particle was at its highest velocity (e.g., June 20 vs. June 24 of 1983). Figure 4 also suggests a small contribution
of base flow (almost zero from early spring to end of June) to observed stream flow; this justifies the calibration of the overland
flow model with the observed stream flow at the Beaverhill Creek stream flow monitoring station.
**3.3 Wetland connectivity**
**3.3.1 Subsurface connections**
The subsurface connectivity map (Figure 5a) indicates that recharge wetlands of a wide range of distances from the
North Saskatchewan River can be connected to the river (red lines). These wetlands range from those located in the moraine,
where the length of connectivity to the river is up to 30 km, as well as those located in the vicinity of the river, where the length
of connectivity to the river is less than 5 km. The total groundwater contribution of these recharge wetlands to the North
Saskatchewan River is 0.775 x $10^6$ m$^3$ per month. Furthermore, water particles released from the recharge wetlands located in the
moraine traverse from hundreds of meters (as small as 100 m) and reach discharge wetlands located in the moraine, or tens of
kilometers (up to 36 km) and reach Beaverhill lake as well as discharge wetlands located outside of the moraine (blue lines).
There is also possibility for subsurface connections between recharge wetlands located at the east of the watershed to Beaverhill
Lake (data not shown).
**3.3.2 Surface connections**
The North Saskatchewan River receives a majority of its wetland-originated surface waters from wetlands located in the
riparian area of the river. For the period from April 1 to August 1 2013, when the largest net surface water fluxes since 2000
occurred, the length of the surface connections to the river ranged from 50 m to 8 km (Figure 6b), with the total surface water
flow contribution from these wetlands being 1.43 x $10^6$ m$^3$ per month. For the period from April 1 to August 1 2009, when the
smallest net surface water fluxes since 2000 occurred, the length of the surface connections to the river ranged from 50 m to 3
km (Figure 6c), with the total surface water flow contribution from these wetlands being 0.81 x $10^6$ m$^3$ per month.
The surface connections among wetlands within the moraine are primarily between neighboring wetlands. For the
period from April 1 to August 1 2013, the length of connection ranged from 25 m to 7 km (Figure 6b); only one water particle
released from wetlands in the moraine reached outside the moraine during this period. For the period from April 1 to August 1
2009, the length of connection ranged from 25 m to 3 km (Figure 6c); no water particle released from wetlands in the moraine
reached outside the moraine during this period.
The modeling approach we used was a one-way linking of subsurface and surface flow processes that could not
consider thoroughly the subsurface flow exfiltration feedbacks on surface flow routing. This simplification had negligible effects
on wetland connectivity within the moraine, as the moraine mostly consists of recharge zones with minimum subsurface
exfiltration (Figure 3). This simplification could have affected the map of surface connectivity of wetlands located in close
vicinity of the North Saskatchewan River (within a 1000 m buffer) wherein subsurface water can exfiltrate and enhance surface
connectivity. However, the rate of subsurface exfiltration in these riparian areas was on average 1 x $10^{-4}$ m/d (Figure 3a), which
is considerably smaller than the average of net atmospheric inputs (precipitation-evapotranspiration) from April 1 to August 1
2013 that was equal to 7 x $10^{-3}$ m/d and from April 1 to August 1 2009 that was equal to 4 x $10^{-3}$ m/d. Therefore, while it is not
known if this one-way linking simplification had negligible effects, it is unlikely that it did particularly during overland flow
events.

### 3.3.3 Timing and length of subsurface and surface connections

Subsurface connections between wetlands and the North Saskatchewan River (Figure 6a) and from wetlands located in moraine to other wetlands in the watershed (Figure 6b) showed a significantly slower and longer time scale compared to surface connections. The average time of the subsurface connections between wetlands and North Saskatchewan River was orders of magnitude longer than the average time of the surface connections. This difference between surface and subsurface transit times led to discontinuity in the continuum of transit times to the river as shown in Figs. 6a and 6b (left panel). Similarly, the average length of subsurface connections of wetlands to the North Saskatchewan River was longer than the average length of surface connections. Furthermore, among wetlands, the average of both time and length of subsurface connections were longer compared to surface connections. These discontinuities between subsurface and surface transit times (and flow lengths) may have been attributed to the lack of characterization of fast subsurface flow in our landscape and our model. Fast subsurface flow has been widely observed in humid forested landscapes with a high frequency of macropores and large hydraulic conductivities in the top tens of centimeters portion of the soil. Available observations in other parts of Canadian Prairie Pothole region support such large shallow hydraulic conductivity (van der Kamp and Hayashi, 2009), which can lead to local fast subsurface flow connectivity among very proximal hydrologic features (as shown using small-scale pond water budget calculation approach by Brannen et al. (2015)). However, we neither have access to the observations that suggest the existence of such large shallow hydraulic conductivity in our part of the Canadian Prairie Pothole region, nor observations that we can calibrate a more complex watershed-scale model which includes such small-scale heterogeneity. Inclusion of such conductive zones in our model (if such zones exist) could decrease subsurface mean transit time and mean flow length and fill the gap between surface and subsurface transit time distributions.

Figure 7 shows the relation between simulated subsurface mean transit time of particles discharged into each discharge wetland and observed concentration and values of Ca, Mg, EC and TDS of the discharge wetland. We expected that the concentration of weathering-derived products and TDS and the value of EC within discharge wetlands would be positively correlated to the subsurface mean transit time of water particles discharged into the wetland. Figure 7 shows a strong positive correlation between simulated mean transit time and the concentration or value of the different constituents within the discharge wetlands. This suggests that our simulated subsurface connectivity map and transit time are valid.

### 3.3.4 Relation between wetland flow contribution to the river and wetland-river distance

To determine whether proximity of wetlands to the North Saskatchewan River is a proxy for their hydrologic connection to the river, we calculated subsurface and surface flow contributions of each wetland to the North Saskatchewan River for a period of four months. The flow contribution-distance relationship of connected wetlands to the river (Figure 8-Left panel) suggests that subsurface flow contribution of wetlands to the river is in general insensitive to their distance to the river, with a negligible correlation coefficient ($\rho = -0.001\%$) between subsurface flow contribution and distance. However, proximal wetlands contribute more surface flow to the river than distal wetlands, with a more pronounced correlation coefficient ($\rho = -15\%$) from April 1 to August 2009, when the smallest net surface water fluxes since 2000 occurred, compared to April 1 to August 2013 ($\rho = -6\%$), when the largest net surface water fluxes since 2000 occurred. Figure 8 (right panel) also shows that the transit time of each wetland connection to the river increases as wetland distance to the river increases; this is true for both surface and subsurface connections.

**3.4 Extendibility to the entire PPR**

2         Figure 9 shows that the distributions of observed shortest distance of wetlands to their nearest wetland neighbor and

shortest distance of wetlands to nearest major stream are statistically similar between Beaverhill watershed and a large portion of
prairie potholes in North America (the theoretical lines between distributions in Quantile-Quantile plot is $x = y$). Table 2a
compares average monthly climatic measures in the Beaverhill watershed to the entire PPR. There is no significant difference in
the median and variance of temperature between Beaverhill watershed and the entire PPR at significance levels of 0.05 and 0.10,
respectively. There is no significant difference in the median and variance of precipitation minus evapotranspiration between the
Beaverhill watershed and the entire PPR ($p > 0.10$). While there is a difference in the median snow water equivalent between
Beaverhill watershed and the entire PPR ($p < 0.10$), there is no significant difference in the variance of SWE ($p = 0.92$). The
geology of the Beaverhill watershed is also consistent with the geology of the entire PPR (Table 2b). These similarities may
suggest that the behavior of subsurface and surface connections of wetlands within the Beaverhill watershed can be extended to
the other parts of the PPR.
**4 Discussion**

14         Hydrologic connectivity of wetlands determines in part their hydrologic, biogeochemical and biological functions.

Hydrologic connectivity, however, is poorly understood and modeled (Cohen et al., 2016). While existing models may be able to
emulate aggregate influence of wetland connectivity on the quantity and quality of downstream waters (e.g., Shook et al., 2013),
very few of these models have been designed with connectivity in mind, and thus are not able to determine the local and regional
interactions between wetlands and other hydrologic features in wetland-dominated landscapes.

19         Here, we couple a steady-state groundwater-surface water interaction model with a transient surface flow routing model

to assess wetland connectivity in a large wetland-dominated watershed within the Prairie Pothole Region. The modeming
approach uses a one-way linking of subsurface and surface flow processes which ignores the potential subsurface exfiltration
feedback on surface flow routing. Nonetheless, the modeling approach enables answers to long-standing questions on watershed-
scale surface and subsurface connections of wetlands that far outweigh its limitations.
*Model performance*

25         The calibrated parameters of the groundwater-surface water interaction model including saturated hydraulic

conductivities of the subsurface and recharge rate were consistent with observations within or close to the Beaverhill watershed.
The groundwater-surface water interaction model predicts reasonably groundwater discharge/recharge areas along the land
surface compared to groundwater discharge/recharge areas inferred from hydraulic head measurements, chemical tracers and
isotopic signatures. The "fill and spill" overland flow model predicted observed stream flow close to the Beaverhill watershed
outlet with acceptable accuracy.
*Timing and length of connection*

32         The model was used to map wetland connectivity and quantify the continuum of time and length variations of this

connectivity. Our results reveal that wetlands in the Beaverhill watershed, with a high density of geographically isolated
wetlands (GIWs), are not hydrologically isolated. Furthermore, the subsurface and surface connections show diverse number,
timing and length. The number of wetlands connected to the major drainage network (here the North Saskatchewan River) from
subsurface pathways was significantly larger than the number of wetlands connected from surface pathways, even in response to

large precipitation events (Figure 5). Fast surface connections originated from the wetlands located near the river (with a maximum distance of 8 km) whereas slow subsurface connections originated from a wide range of close and distant wetlands with a maximum distance of 30 km from the river. Indeed, model simulations reveal that regional surface waters integrate a wide range of continuum of time and length variations of connectivity, not just rapid or surface-connected flowpaths located at the top of this continuum.

Watershed hydrologic, biogeochemical and biological functions in wetland-dominated landscapes such as the Beaverhill watershed are influenced by the transit times, velocities (rates) and mode (pathways) of hydrologic connection (Bracken and Croke, 2007;Cohen et al., 2016). Wetlands that connect rapidly (but not persistently) to the river via surface fill-and-spill mechanism constrain peak flow volumes, delay peak timing, and retain sediments (Craft and Casey, 2000). However, wetlands that connect to the river only via slower subsurface flowpaths regulate water table depth (Lane et al., 2015), maintain base flow and recession rate of river hydrographs (McLaughlin et al., 2014;Golden et al., 2015), and retain and transform pollutants (Marton et al., 2015). Our results show that as distance between wetlands and the North Saskatchewan River increases, connection time between wetlands and the river also increases (Figure 8).  Therefore, while the flow contribution from surface and subsurface connections is similar, the time-scales at which the subsurface connections travel from source to river is much longer (i.e., the subsurface flow contribution represents water that may be much older).  The long transit times of subsurface connections of distal wetlands may impact significantly biogeochemical processes in the vicinity of the river, improving water quality of regional water resources. For example, regional subsurface pathlines originated from distal wetlands can facilitate the completion of kinetically-limited reactions and enhance retention, sorption, and transformation of sorbing nutrients (Min et al., 2010), metals (Mays and Edwards, 2001) and likely pesticides (Ameli, 2016), all of which positively influence aquatic ecosystem structure and function (e.g., Euliss et al., 2004;Cook and Hauer, 2007). Figure 6c depicts the simulated cumulative probability of transit time distributions of water flowing from wetlands to the North Saskatchewan River (ensemble of surface and subsurface wetland-originated contributions), and summarizes the potential ecosystem services of each portion of this continuum.

*Distance not a proxy for connectivity*

Our results show that all wetlands located within a distance of 30 km can affect the quantity and quality of water in the North Saskatchewan River. Contrary to Hayashi et al. (2016) who hypothesized that wetlands in PPR can only be connected to closely adjacent water bodies and subsurface flow through the deeper portions is insignificant, our findings using novel 3D watershed-scale subsurface-surface connectivity model corroborates the findings of Winter et al. (2003) who showed that distal sources can impact the quality and quantity of water in the regional surface water bodies in Prairie Pothole landscape at North Dakota. Quantification of the contribution of wetlands to the river suggests that slow subsurface flow contribution to the river flow is substantial ($0.775 \times 10^6$ $m^3$ per month) and comparable to the surface flow contribution($0.81 \times 10^6$ $m^3$ per month during the driest year and $1.43 \times 10^6$ $m^3$ per month during the wettest year since 2000). In addition, although the surface flow contribution from wetlands to the river is correlated with the wetlands distance to the river, subsurface flow contribution from wetlands to the river had a weak relationship with the distance between wetlands and the river (Figure 8), and a broad range of proximal and distal wetlands can be connected to the river (Figure 6a) and influence the river quality and quantity. This implies that decisions to protect GIWs based only on distance of the wetland to a river (e.g., 2015 U.S. Clean Water Rule (Federal Register 80: 37054-37127)), may lead to the loss of distal wetlands with important watershed functions.

These findings can be extended to the entire PPR, since the climate, geology and topography can be considered almost (statistically) similar throughout the PPR. Although Figure 9 suggests that the distribution of wetlands does not differ

significantly between the Beaverhill watershed and the Prairie pothole landscape in North Dakota, this Figure shows a higher density of wetlands in the range of long distances among wetlands in North Dakota compared to the Beaverhill watershed (i.e., more distant wetlands are relatively closer to one another in North Dakota). The longer distances among wetlands in the Beaverhill watershed may imply a less frequent surface hydrologic connectivity among wetlands in the Beaverhill watershed compared to the North Dakota landscape.

*Guidelines for wetland protection, removal and restoration*

Human alteration has changed the natural continuum and timing of hydrologic connectivity (Min et al., 2010;Pringle, 2003). Given that the aforementioned ecosystem services accrue from a continuum of transit times, the cumulative impact of such alteration can be significant (Johnston, 1991;Zedler, 2003). Current wetland management strategies in the PPR are likely to lead to loss of wetlands (particularly GIWs) located far from regional surface waters (Van Meter and Basu, 2015;Serran and Creed, 2015). Removing these wetlands can increase surface pathways towards rivers with potential consequences of flooding and eutrophication during large events. For example, the loss of GIWs in watersheds including the Beaverhill watershed has been implicated as one cause of the increase in phosphorus loading to Lake Winnipeg, located 1,300 km east of the Beaverhill watershed, leading to eutrophication events and the 2013 listing of Lake Winnipeg as the most threatened lake in the world (Ulrich et al., 2016).

**5 Conclusion**

A one-way linked subsurface-surface model was developed to assess the continuum of time and distance variations of hydrologic connectivity of wetlands in a large watershed with a high density of geographically-isolated wetlands in the Prairie Pothole Region. The model showed that wetlands are not hydrologically isolated, and that the surface and subsurface hydrologic connections vary significantly in terms of their timing and length. Contributions of slow, subsurface connections from both proximal and distal wetlands to the river are substantial and comparable to the contributions of fast, surface connections. The subsurface-surface model is computationally efficient, enabling upscaling to the entire Prairie Pothole Region (and elsewhere) to assess wetland connectivity that was heretofore difficult to quantify, and providing guidance on the development of watershed management and conservation plans (e.g., wetlands drainage/restoration) under different climate and land management scenarios. Our modeling approach can explicitly assess and evaluate the hydrologic connectivity of individual wetlands, providing scientists and conservation authorities with information to understand and manage the potential response of the entire watershed to direct and indirect changes such as wetland drainage or restoration. We recommend coupling robust hydrologic connectivity models with biogeochemical and biological data can (1) improve our understanding of landscape hydrologic connectivity and its impact on the structure and function of wetlands, and (2) aid in the assessment of feedbacks between hydrology, biogeochemistry and biology.

**Data and code availability**

The data used in this paper are available upon request from the corresponding author.

**Appendix A: Mathematical formulation of Groundwater-Surface water interaction model**
At each layer ($m = 1 \ldots M$) of an unconfined aquifer, the exact 3D series solution to the saturated steady flow
governing equation, with no-flow conditions along the sides of the computational domain, was obtained in terms of discharge
potential function ($\phi_m = K_m h_s$) as (Ameli and Craig, 2014):

$$\phi_m(x, y, z) = \sum_{j=0}^{J-1} \sum_{n=0}^{N-1} \cos \omega_j x \, \cos \omega_n y \, [A_{jn}^m \cosh(\gamma_{jn} z) + B_{jn}^m \sinh(\gamma_{jn} z)] \tag{A.1}$$

where $\quad \omega_j = \frac{j\pi}{L_x}; \, \omega_n = \frac{n\pi}{L_y}; \quad \gamma_{jn} = \pi \sqrt{\frac{j^2}{L_x^2} + \frac{n^2}{L_y^2}} \quad$ for $\quad j = 0 \ldots J - 1 \, \& \, n = 0 \ldots N - 1$
In eq. (A.1), $h_s(x, y, z)$ is the total hydraulic head, and $K_m$ (LT$^{-1}$) is the $m^{\text{th}}$ layer saturated hydraulic conductivity. In addition, $j$
and $n$ represent the coefficient index, whereas $J$ and $N$ refer to number of series in the $x$ and $y$ directions, respectively. A total of
144 series terms ($N$=12 and $J$=12) were used. The series coefficients associated with the $m^{\text{th}}$ layer and $j^{\text{th}}$ and $n^{\text{th}}$ series term are
$A_{jn}^m$ and $B_{jn}^m$. The rectangular computational domain with no-flow side boundaries has a length of $L_x = 84.5$ km and $L_y = 95.4$
km in $x$ and $y$ directions, respectively, which embeds the watershed boundary as shown in Fig. 1. No-flow side boundaries were
placed on average 20 km away from the watershed original border; this treatment minimized the impact of side boundaries on
flow behaviour and subsurface connections. A continuous map of hydraulic head was then obtained as:

$$h_s(x, y, z) = \frac{\phi_m(x, y, z)}{K_m} \tag{A.2}$$

To complete the solution, the unknown series solution coefficients of each layer ($A_{jn}^m$, $B_{jn}^m$) were calculated by imposing
infiltration rate along the topographic surface, the no-flow condition along the bottom boundary, and the continuity of flux and
head conditions along the layer interface of the multi-layer unconfined aquifer. A simple numerical least square scheme was used
to impose these boundary and continuity conditions. In general, this continuous solution (Eq. (A.1)) exactly satisfies the mass
balance and Darcy equations in the entire watershed, except along the boundaries where mass balance and Darcy equations are
prone to numerical least square error. Ameli and Craig (2014) showed that this error can be negligible when sufficient number
of control points was used within numerical least square algorithm. To ensure minimum numerical least square error along the
boundary and layer interfaces, 806130 control points (uniformly-spaced at each 100 m) were placed along two boundary
interfaces and the layer interface of the computational domain.
The continuous maps of Darcy fluxes at the $m^{\text{th}}$ layer and at each $x$, $y$ and $z$ directions can be computed by the following
equation:

$$q_m^x(x, y, z) = \frac{d\phi_m(x, y, z)}{dx} \tag{A.3a}$$

$$q_m^y(x, y, z) = \frac{d\phi_m(x, y, z)}{dy} \tag{A.3b}$$

$$q_m^z(x, y, z) = \frac{d\phi_m(x, y, z)}{dz} \tag{A.3c}$$

Equation (A.3) can also be used to determine groundwater discharge fluxes at discharge areas (seepage faces) along the
land surface as well as groundwater recharge fluxes where the water table is below the land surface. A subsurface map of pore
water velocities ($V$), which is required to perform subsurface water particle tracking, was obtained as:
$$V_m^x(x, y, z) = \frac{1}{\theta_s} q_m^x(x, y, z) \tag{A.4a}$$

$$V_m^y(x, y, z) = \frac{1}{\theta_s} q_m^y(x, y, z) \tag{A.4b}$$

$$V_m^z(x, y, z) = \frac{1}{\theta_s} q_m^z(x, y, z) \tag{A.4c}$$

where $\theta_s$ is subsurface porosity. Inputs to the model included: (1) the location and water level of wetlands; and (2) material
properties of the subsurface. For (1), the delineated wetlands explained in Sect. 2.2 were used, and we assumed that the water

level was equal to the average elevation of each wetland boundary. This water level was used as a constant head boundary condition at each wetland. For (2), a two-layer unconfined aquifer with a 5 m thick shallow layer was used to characterize the subsurface (as suggested in van der Kamp and Hayashi, 2009). The bottom boundary of the computational domain was assumed to be at Z = 0 with a no-flow condition. A porosity ($\theta_s$) value of 0.14 equal to the average measured porosity at the Beaverhill watershed (Gleeson et al., 2014) was also used.

**Competing interests**

The authors declare that they have no conflict of interest.

**Acknowledgements**

This research was funded by a NSERC Discovery Grant and an Alberta Land Institute Grant to IFC. The authors gratefully acknowledge Alberta Agriculture and Forestry, Alberta Environment and Parks, Alberta Geological Survey and Environment and Climate Change Canada for providing some of the data and measurements used in this paper. The authors would also like to thank Dr. Jeffrey McDonnell, Dr. Allan Rodhe, Dr. Kevin Bishop and Dr. Thomas Grabs for providing insightful comments which improved the quality of the manuscript.

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

**Table 1: Average concentration of chemical tracers in hydrologically-simulated discharge and recharge areas, as well as p-values of non-parametric Wilcoxon rank sum test (equality of median) used to assess the differences between simulated groundwater discharge and recharge areas. p-values less than 0.10 depict a statistically significant deference between chemistry of simulated groundwater discharge and recharge areas. a) Concentration of chemical tracers in hydrologically-simulated groundwater discharge and recharge areas, b) concentration of chemical and isotopic tracers in hydrologically-simulated discharge and recharge wetlands. The reported values in the parenthesis for $^{18}$O (and $^{2}$H) are the relative difference between average isotopic concentrations (ratio) in simulated discharge or recharge wetlands from average watershed concentration.**

a)

|  | Ca(mg/l) | Mg(mg/l) | TDS(mg/l) | EC($\mu$S/cm) |
|---|---|---|---|---|
| Discharge | 123 | 46 | 1189 | 1631 |
| Recharge | 102 | 43 | 995 | 1443 |
| *p*-value | 0.09 | 0.02 | 0.09 | 0.05 |

b)

|  | Ca(mg/l) | Mg(mg/l) | TDS(mg/l) | EC($\mu$S/cm) | ALK(mEq/l) | $^{18}$O (‰) | $^{2}$H (‰) |
|---|---|---|---|---|---|---|---|
| Discharge | 65 | 42 | 1144 | 1502 | 241 | -8.90 (3) | -105.37 (8) |
| Recharge | 59 | 34 | 721 | 923 | 195 | -6.28 (0.9) | -91.55 (3) |
| *p*-value | 0.08 | 0.14 | 0.07 | 0.06 | 0.07 | <0.001 | <0.001 |

**Table 2: Comparison of climate and geological features between Beaverhill watershed and the entire PPR. a) p-values of Wilcoxon rank sum and Levene tests that were used to assess the similarities in median and variance, respectively, between monthly-averaged climatic measures in the Beaverhill watershed and the entire Prairie Pothole Region. These statistical analyses were conducted based on 31 years (from 1981 to 2011) precipitation minus actual evapotranspiration (P-ET), snow water equivalent (SWE) and air temperature data. p-values greater than 0.10 indicate a similarity between climatic measures at a significance level of 0.10. b) Comparison of the average values of geological features between Beaverhill watershed and the entire PPR. K (m$^2$) refers to permeability.**

a)

|  | Wilcoxon rank sum | Levene |
|---|---|---|
| P-ET | 0.38 | 0.44 |
| SWE | 0.02 | 0.92 |
| Temperature | 0.08 | 0.19 |

b)

|  | PPR | Beaverhill |
|---|---|---|
| Mean Porosity (%) | 0.15 | 0.14 |
| Mean Permeability No Permafrost (log(k)) | -14.94 | -15.13 |
| Mean Permeability Permafrost (log(k)) | -14.94 | -15.13 |
| Mean Permeability Standard Deviation (m$^2$) | 1.79 | 1.82 |
| Bedrock Geology - Sedimentary Rocks (% Area) | 96.90% | 97.67% |

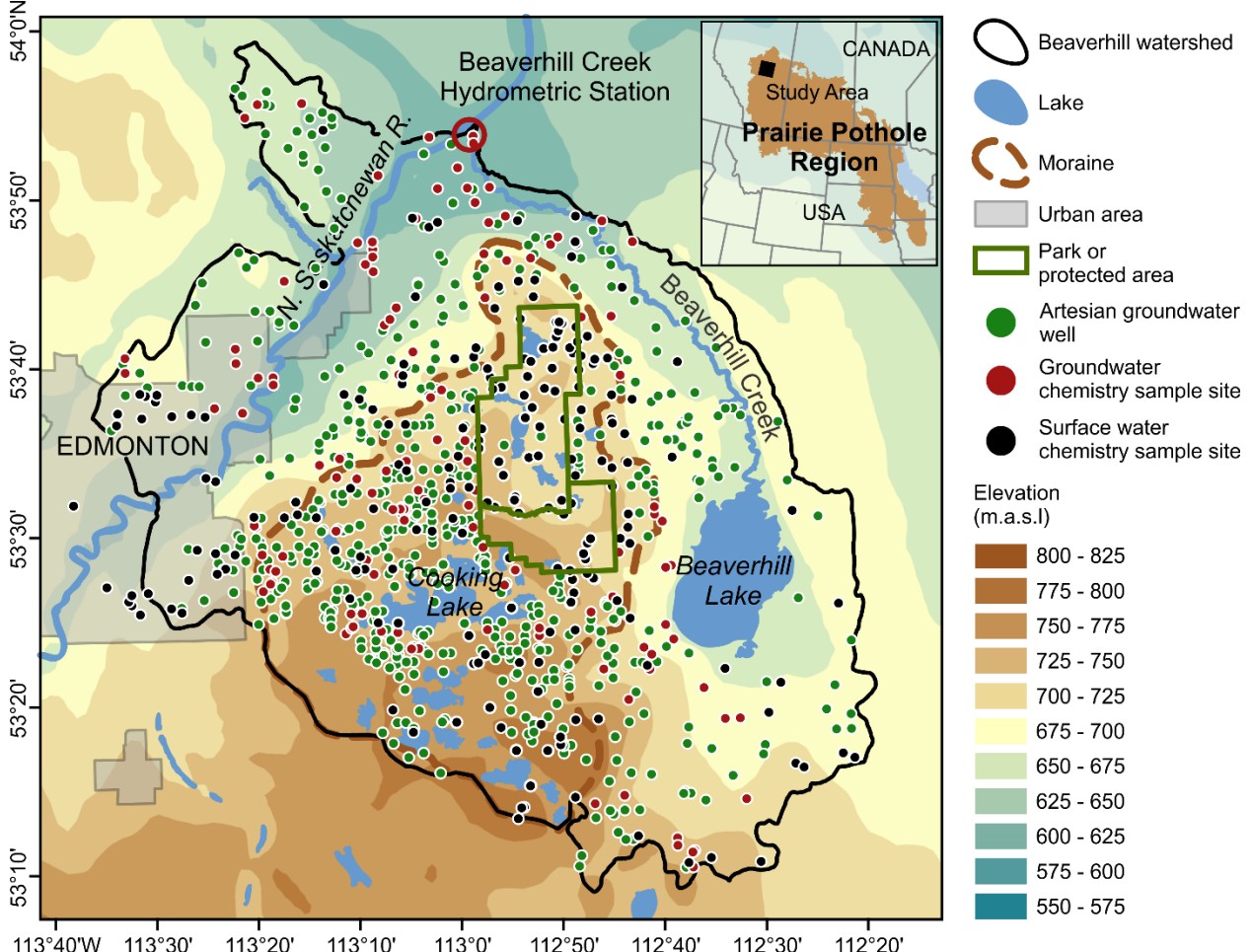

**Figure 1: Beaverhill watershed, Alberta, Canada. The location of 1,413 artesian groundwater wells installed in the**
**bedrock and screened 30 to 80 m below the land surface are shown (green dots). Black dots depict the location of 208**
**lakes, wetlands and ponds wherein chemistry and isotopic measurements were taken. Red dots depict the location of 121**
**shallow (< 10 m deep) groundwater wells wherein groundwater chemistry measurements were taken. The red ring shows**
**the location of Beaverhill Creek monitoring station. The inset shows the map of the Prairie Pothole region of North**
**America.**

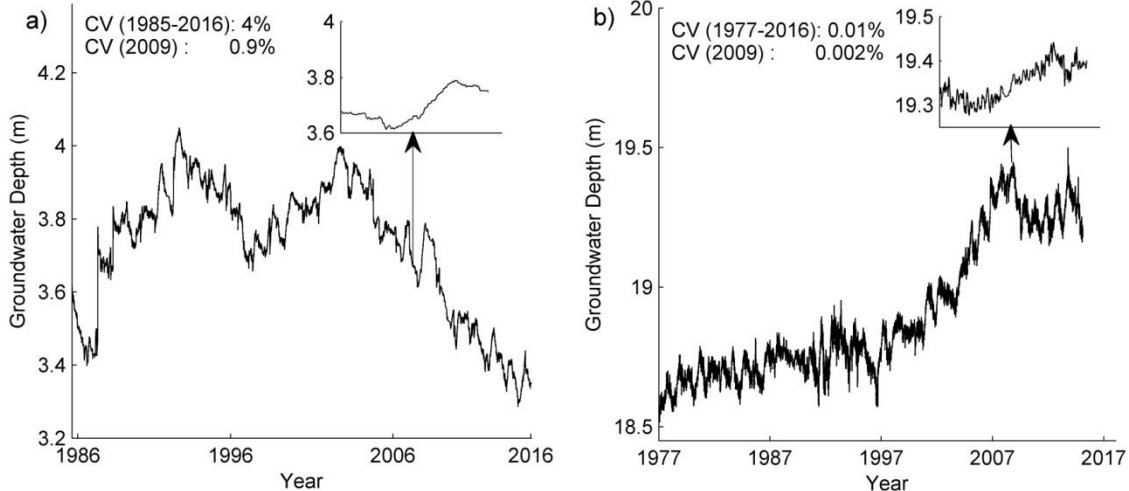

**Figure 2: Observed time series of groundwater depth at two measurement stations located east and west of the Beaverhill watershed. a) Groundwater depth from August 1985 to July 2016 at the Vegreville measurement station located 15 km east of the Beaverhill watershed boundary. b) Groundwater depth from October 1977 to October 2016 at the Barrhead measurement station located 65 km east of the Beaverhill watershed boundary. The insets show the time series of groundwater depth in 2009, when the observations used here to develop the steady-state groundwater-surface water interaction model were available. CV refers to the coefficient of variation of groundwater depth data during the given period.**

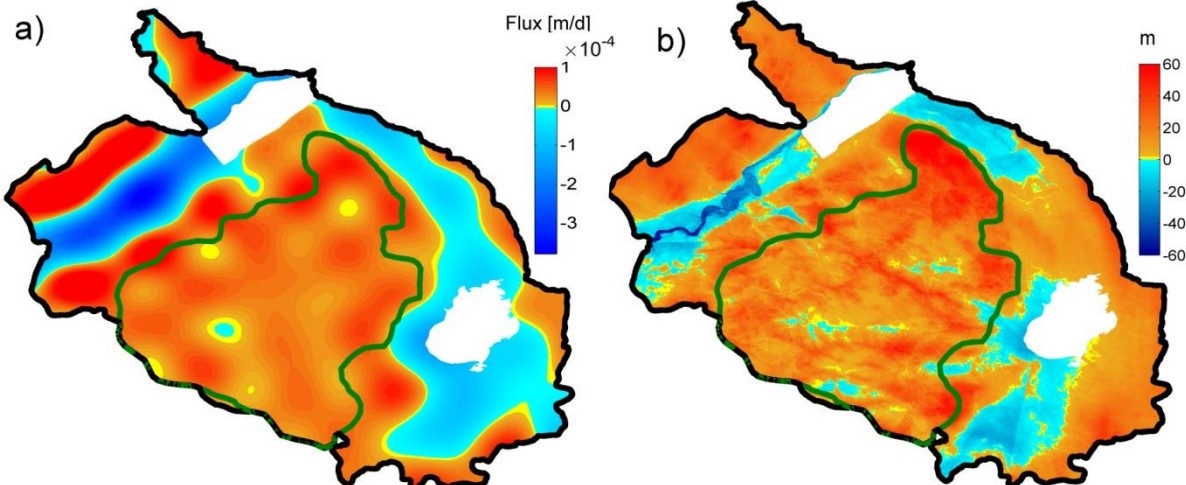

**Figure 3: Comparison between simulated and inferred groundwater discharge/recharge areas. (a) Simulated**
**groundwater discharge (blue surfaces with negative groundwater fluxes) and recharge (red surfaces with positive**
**groundwater fluxes) areas. (b) Inferred groundwater discharge (blue surfaces) and recharge (red surfaces) areas from**
**the potentiometric surface generated using measurements from 1,413 artesian wells. Areas where the presences of the**
**Artesian wells were sparse (i.e., at the Beaverhill lake and in the vicinity of North Saskatchewan River, see Figure 1) were**
**extracted. The correlation coefficient between simulated groundwater fluxes at the land surface and the distance of**
**potentiometric surface above and below land surface is 75% (p < 0.001).**

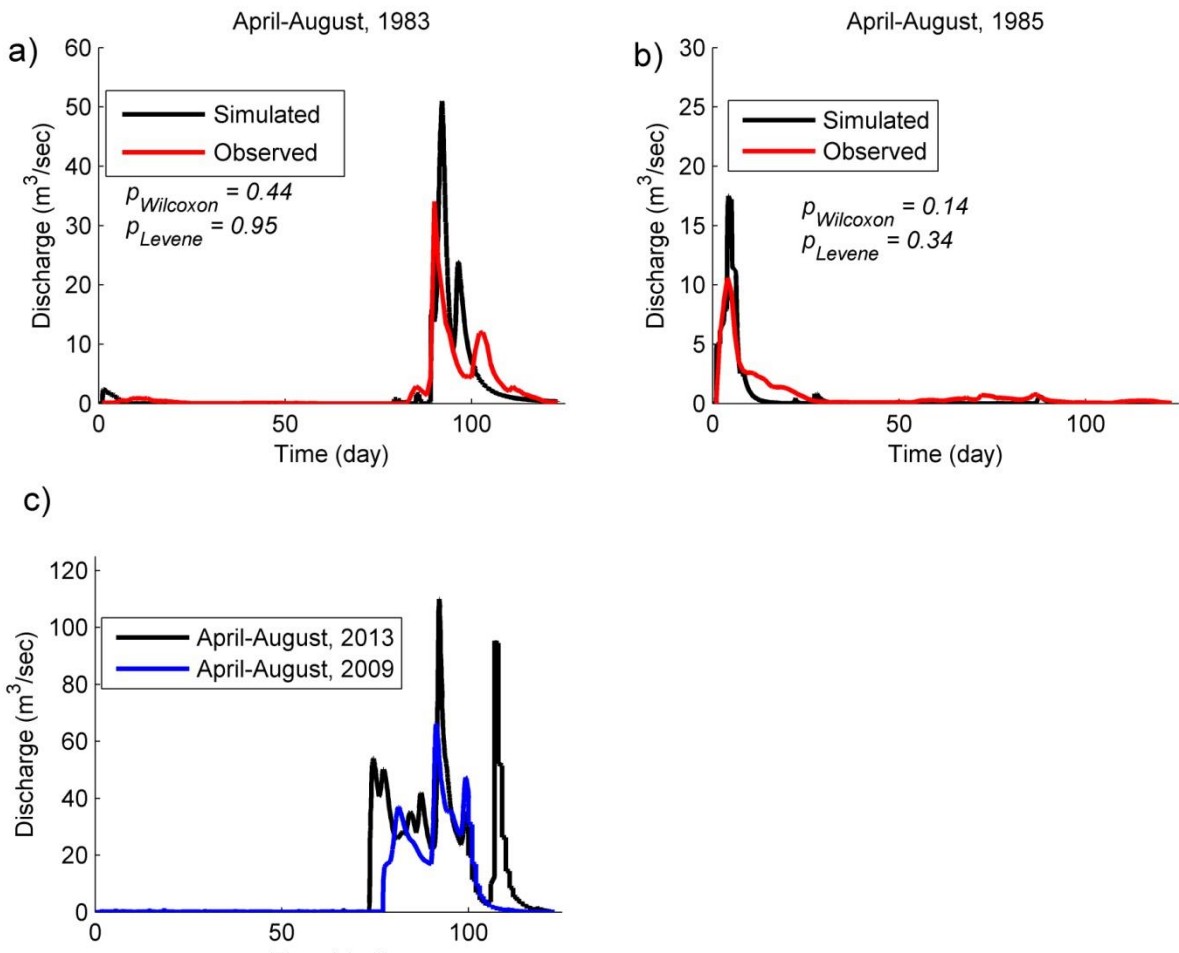

**Figure 4: Simulated and observed stream flow at the Beaverhill Creek monitoring station. a) Performance of the**
**calibrated overland flow model in the simulation of stream flow against observed stream flow for the wettest period when**
**the observed stream flow measurements were available from April 1 to August 1, 1983. The correlation coefficient**
**between observed and simulated hydrographs was 87%(p < 0.001).  b) Performance of the overland flow model in the**
**validation phase for the driest period when the observed stream flow measurements were available from April 1 to**
**August 1, 1985. The correlation coefficient between observed and simulated hydrographs was 45%(p < 0.001). P_wilcoxon**
**and P_Levene refer to the P values of Wilcoxon and Levene tests used to assess the similarity in median and variance**
**between two hydrographs. c) Simulated hydrograph from April 1 to August 1, 2013 when the largest net surface water**
**fluxes since 2000 occurred, and from April 1 to August 1, 2009 when the smallest net surface water fluxes since 2000**
**occurred.**

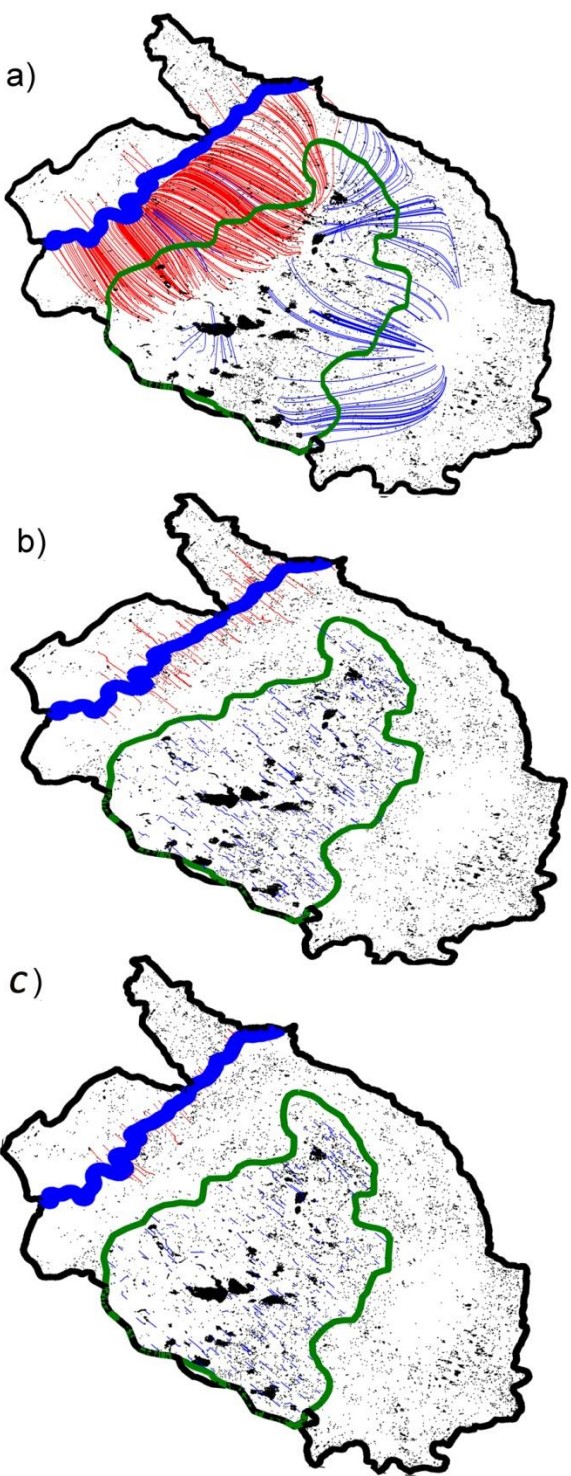

**Figure 5: Hydrologic connectivity among wetlands (blue lines) and between wetlands and North Saskatchewan River (red lines). a) Map of subsurface connections, only particles released from recharge wetlands located in the moraine and reached the Beaverhill lake and discharge wetlands (blue lines), and particles discharged into North Saskatchewan River from recharge wetlands (red lines) are shown. b) Map of surface connections for the period from April 1 to August 2013, when the largest net surface water fluxes since 2000 occurred. c) Map of surface connections for the period from April 1 to August 2009, when the smallest net surface water fluxes since 2000 occurred. Only surface connections between wetlands and North Saskatchewan River (red lines), and connections among wetlands within the Moraine (blue lines) are shown.**

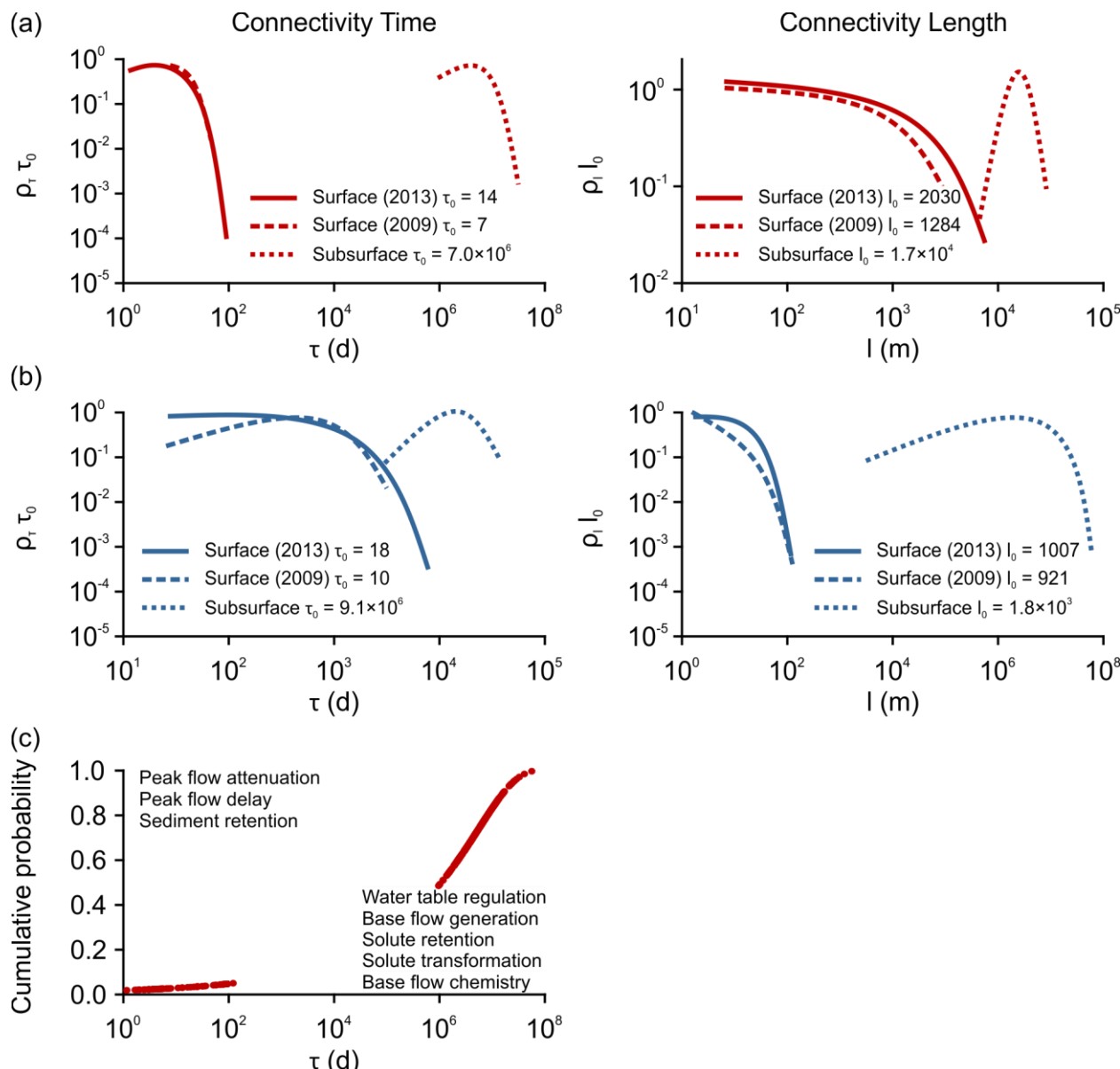

Figure 6: Fitted probability density function of subsurface or surface (both 2009 and 2013) connection transit times ($\rho_\tau$-
left panel) and connection lengths ($\rho_l$-right panel) a) between wetlands and North Saskatchewan River, and b) from
wetlands located in the moraine and other wetlands throughout the watershed. $l_0$ [m] and $\tau_0$ [d] refer to the average
length and transit time, respectively. The axes labels were explained in section 2.3.3 (Equation 2). c) Cumulative
probability of transit time distribution of water particles discharged from wetlands into North Saskatchewan River. The
potential ecosystem services of each portion were also shown.

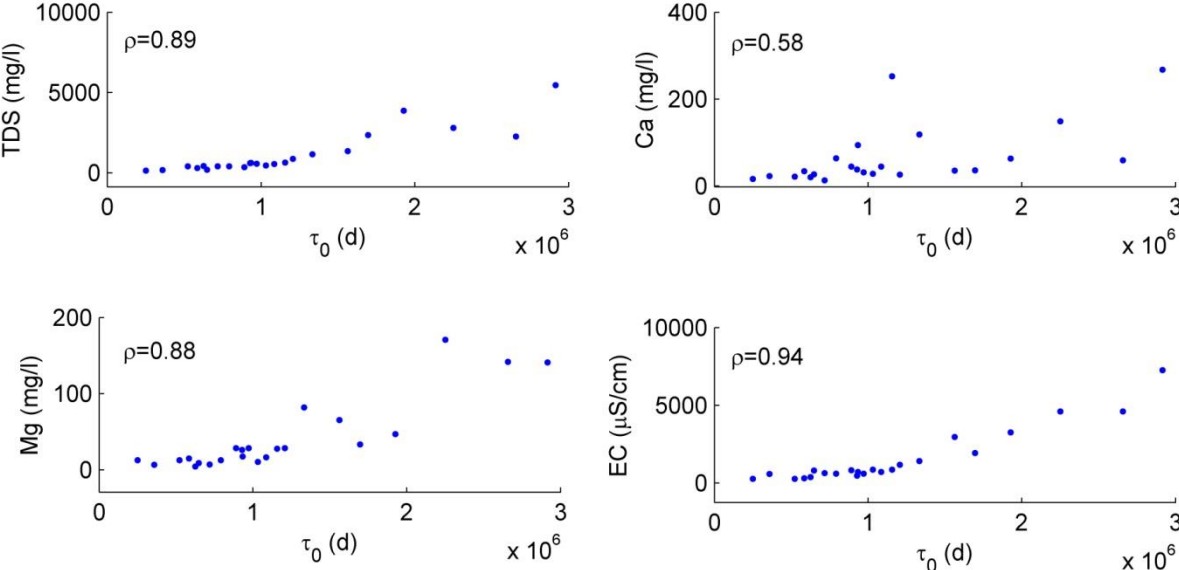

**Figure 7: Relation between simulated subsurface mean transit time ($\tau_0$) of each discharge wetland and the concentration**
**of various chemical constituents in the wetland. Here, the pathlines discharged into each wetland and their associated**
**transit times were calculated by back tracking from 100 uniformly-distributed particle release points located at each**
**discharge wetland. $\rho$ refers to the correlation coefficient between $\tau_0$ chemical concentrations.**

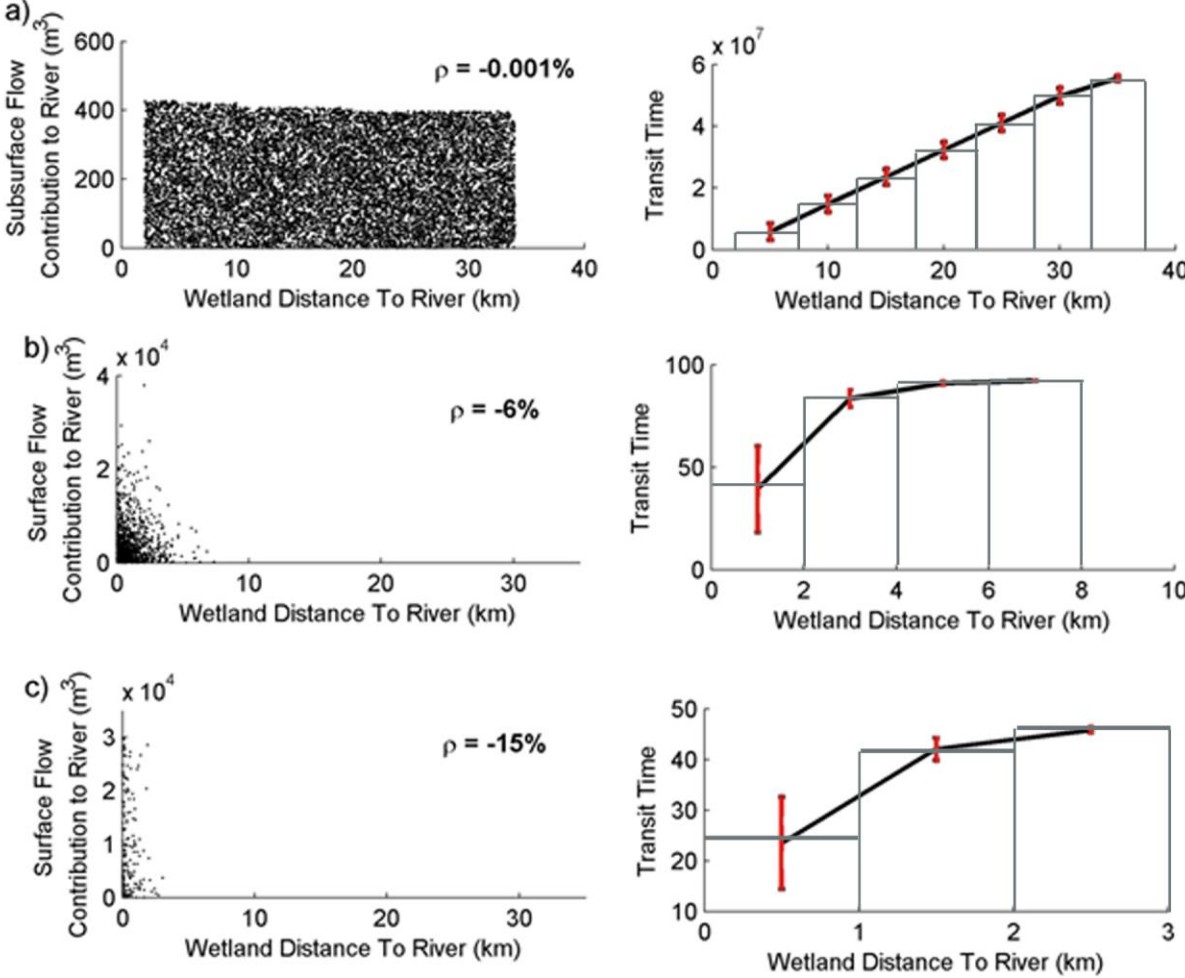

Figure 8 .The relationship between flow contribution of each wetland to the North Saskatchewan River and the distance of the wetland to the North Saskatchewan River (left panel), and the relationship between transit time of connection of each wetland to the North Saskatchewan River and the distance of the wetland to the North Saskatchewan River (right panel). In the right panel the black lines connect the mean value of each distance category and red vertical lines show the standard deviation of each category. a) Subsurface flow contribution and connection of each wetland. Flow contribution was calculated for four months for each wetland using the steady-state model. b) Surface flow contribution in 2013 and connection of each wetland. Flow contribution was calculated for four months (April 1 to August 2013) using the transient model. c) Surface flow contribution in 2009 and connection of each wetland. Flow contribution was calculated for four months (April 1 to August 2009) using the transient model.

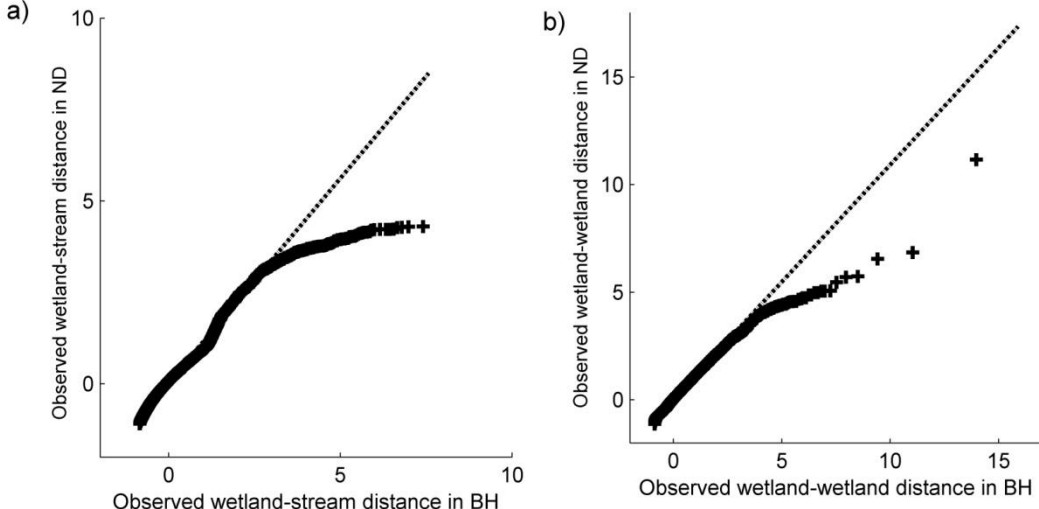

**Figure 9: Quantile-Quantile (Q-Q) plot comparing the distributions of observed shortest distances between surface water**
**bodies in Beaverhill watershed (BH) and prairie potholes in North Dakota (ND). (a) Standardized wetland-stream**
**distance (observed shortest distances of wetland to nearest major stream) in the Beaverhill watershed (BH) vs.**
**standardized wetland-stream distance in prairie potholes in North Dakota (ND). (b) Standardized wetland-wetland**
**distance (observed shortest distances of wetlands to their nearest wetland neighbor) in the Beaverhill watershed (BH) vs.**
**standardized observed wetland-wetland distance in prairie potholes in North Dakota (ND).**

