# Peer review of "Quantifying hydrologic connectivity of wetlands to surface water"

_Hydrology and Earth System Sciences, 2016_

## Referee Comment (RC1) · Anonymous Referee #1 · 10 Sep 2016

This paper covers a very timely topic and would be a nice addition to HESS. The concept of hydrological connectivity is still in its infancy, but its relevance to the wetland management is obvious, even as hydrologists are still learning how to apply the concept. The authors are to be commended on their efforts to advance the thinking on this subject. The study summarized in this paper applies a series of process based models to quantify surface and subsurface hydrologic connectivity among wetlands and a major river, in order to address several goals. These include assessing the performance of the models, comparing the relative importance of surface and subsurface connections, determining if proximity can be used as a substitute for connectivity, and if their findings could be extrapolated beyond the study watershed. The authors meet all these goals but only to different degrees, and I have provided some suggestions that might elevate the study and manuscript. There are some major comments, and numerous

minor ones.

MAJOR COMMENTS

1) Could the authors perhaps present data from the surface overland flow model for a dry year? I understand why they selected 2013, but it would be good to know that the model could represent a condition that is drier, and what those repercussions are for connectivity. One downside of the research as presented is, it does not necessarily present the spectrum of connectivity that could occur in the Beaverhill watershed.

2) A more critical assessment of the simulated surface flow hydrograph is needed. The high regression coefficient is likely because of the low flow period, and the spring peak, which is relatively well simulated. The true test of a modeled surface stream hydrograph in the Prairie Pothole Region is how well it represents the summer recession, any summer events, and timing of the cessation of streamflow. The model does not do this particularly well. The manuscript would be improved if the authors explain their theories as to why the model simulated an event that did not happen, and missed one that did. Could it be that the model missed some important re-connection? If so, why? This will help inform how the model is behaving and provide some great insight.

3) I would argue that the authors misinterpret the content of Figure 9. There is good fit for short distances, but not long. Could the authors please provide more information on how the shortest distances were calculated? Are these Euclidian (ie "as the crow flies") estimates? Or are they along the topographic flow path? Did they come from the digital elevation model? If this is the case, this might explain the departure from the linear function in Figure 9. If I interpret the results correctly, this highlights the problem with the variety of connectivity metrics, measures and indices that are currently used in hydrology. To really address their goal of determining if proximity is a substitute for connectivity, it would be great if the authors could output the contribution of flow from each wetland to the North Saskatchewan River, and plot these flows against distance. This would truly show if distance is (or is not) a proxy for connectivity. The authors do
not use a metric that demonstrates the magnitude of connectivity, only its presence or absence. They need one for magnitude to answer their question if proximity can be used as a substitute for connectivity.

MINOR COMMENTS

Some relevant work the authors should consider working into the manuscript are listed below.

Shook, K., J.W. Pomeroy, C. Spence and L. Boychuk, 2013. Storage dynamics simulations in prairie wetlands hydrology models: evaluation and parameterization, Hydrological Processes 27: 1875 – 1889.

Brannen, R. C. Spence and A. Ireson, 2015. Influence of shallow groundwater-surface water interactions on the hydrological connectivity and water budget of a wetland complex, Hydrological Processes 29: 3862-3877.

Hayashi, M., G. van der Kamp and D. Rosenberry, 2016. Hydrology of prairie wetlands: understanding the integrated surface-water and groundwater processes, Wetlands doi: 10.1007/s13157-016-0797-9

Page 1 Line 22: Could read: " ….. protection, as these are small features typically vulnerable to drainage or manipulation ……" As for the rest of the sentence, please provide information on why being numerous equates to a need for protection.

Page 1 Line 25: Maybe reference Brannen et al. here too.

Page 1 Line 26: I know that fill-and-spill has become common vernacular, but perhaps the authors could say "…… via mechanisms analogous to fill-and-spill runoff generation (Rains et al., 2006)."

Page 1 Line 29: Be very careful when using the term "function" because it has very specific meanings depending on the context. For instance, the hydrological function of a specific wetland using the hydrogeomorphic assessment method, which can be

required for development works, follows methodologies necessary for the specific purpose of discerning a loss or gain in wetland function relative to a reference standard. This approach was designed to detect and measure variation in function due to human impacts, not natural variation. In contrast, Black (1997) proposed that landscape units have hydrologic functions such as collecting, storing and discharging. Could I suggest the authors explicitly define what they mean by "function"? Or, use the word to "role".

Page 2 Line 6: Perhaps instead of committing to a statement that an inability to quantify connectivity would lead to preferential protection to certain types of wetlands, maybe say " …. may lead to incorrect or inappropriate management decisions regarding wetland removal, protection or reclamation."

Page 3 Line 11: remove italics here and throughout this section.

Page 3 Line 12: Maybe provide a URL for the climate data. Page 3 Line 15: Maybe rephrase to: ….although snowmelt can be an important to runoff in the spring."

Page 3 Line 34: Do the authors mean the probability of depression existence or presence?

Page 3 Line 37: What are "integrated wetland features"?

Page 4 Line 1: In recent years in the Prairie Pothole Region what would normally be considered GIWs had ponds that have been above their surface outlet elevations. Perhaps a sentence or two would be a good idea on how often a GIW needs to be not spilling in order to be considered a GIW.

Page 5 Line 33: Please explain why there is such a short calibration period. The gauge was open until 1986.

Page 6 Line 7: Just my preference, but more detail in the paper on the methods would be helpful for the reader, particularly the water particle tracking approach and how surface water velocities were approximated. Page 6 Line 22: Could I suggest the Hayashi paper I note above be worked into the context here? Hayashi and his co-authors

present a new conceptual model of subsurface flow in the Prairie Pothole Region that is a major departure from the model of Toth that is the basis for the assumption that geographic proximity is an indicator of connectivity.

Page 6 Line 29: Maybe rephrase to: ". . ..will be linear but not following y=x."

Page 6 Line 31: Please rearrange this sentence.

Results: The description of the results reads a bit terse. Sometimes the content seems little more than a figure caption. Could I suggest the authors provide more description on the results, particularly where the model does not work well.

Figure 9: It is unclear where the North Dakota data are from. Could the authors provide this detail in the Methods section.

Page 9 Line 15: Maybe discuss within the context of the results of Shook et al.

Page 9 Line 36: Figure 6 does not illustrate what is discussed here.

Figure 10: The authors need a more explicit explanation of how they decided which services were associated with each portion of this curve.

Conclusions: Just a comment, but even though most of the hydrology community knows that wetlands are not hydrologically isolated, I completely agree that it is good to make this point.

Table 1: Is the p value for magnesium correct? It seems small, especially in light of the content of Table 2.

Figure 2: The last word in the caption "time", could be "period".

Figure 3: Great figure.

Figure 4: It is hard to see the wetlands in this. If this figure was created by clipping Figure 3 by a wetland layer, my suggestion is that you delete Figure 4 because it does not add too much information.

Figure 10: Why is there a gap?

[Figure]

---

## Referee Comment (RC2) · Anonymous Referee #2 · 11 Oct 2016

Authors characterized surface water and subsurface connectivity of wetlands using a physically based surface-subsurface model. Groundwater level measurements, water chemistry and stable water isotopes are used to illustrate the model performance at recharge and discharge locations. While this is an interesting study, the study can benefit by providing more quantitative measures of model performance compared to observations, justification of the modelling approach compared to the existing coupled surface water-subsurface models and sensitivity analysis.

1) Authors should provide a more quantitative measure of model performance. For example in Figure 3, authors qualitatively compare simulated recharge/discharge areas with interpolated groundwater observations. Similarly, water quality data are used to indicate differences between recharge and discharge zones using the Wilcoxon rank sum test. In Figure 5, the model predicts the second peak much earlier than the observations. The paper can greatly benefit by providing further details about the model's performance as well as discussions about discrepancy observed between simulated and observed outputs.

2) Authors have used a grid-free subsurface flow model to simulate groundwater flow and then used the 2D transient surface water flow of HydroGeosphere. It is not clear why authors did not use HydroGeosphere in the first place as it provides an integrated system to simulate surface water-groundwater interactions. I understand that the grid-free approach is computationally more efficient but authors should justify their approach. Indeed it would be really interesting to see how HydroGeosphere simulations compare with the modelling approach that authors developed. How much loss in accuracy is obtained by assuming steady state groundwater condition in the grid free approach compared to transient simulations?

3) How does the "semi-coupling" approach of surface-subsurface processes in the model impact capturing wetland connectivity and travel time distributions? Moreover, would it be more suitable to use the term one-way coupling instead of semi-coupling as the feedback from the subsurface is not included in this approach?

4) It will be interesting to investigate how changes in climatic condition impact wetland connectivity and travel time distributions.

5) It will be useful if authors provide further details about the model input and time step.

6) Authors need to provide further details about the calibration approach and identify the performance of the model for calibration and evaluation periods.

---

## Author Comment (AC1) · 7 Nov 2016

COMMENT: This paper covers a very timely topic and would be a nice addition to HESS. The concept of hydrological connectivity is still in its infancy, but its relevance to the wetland management is obvious, even as hydrologists are still learning how to apply the concept. The authors are to be commended on their efforts to advance the thinking on this subject. The study summarized in this paper applies a series of process based models to quantify surface and subsurface hydrologic connectivity among wetlands and a major river, in order to address several goals. These include assessing the performance of the models, comparing the relative importance of surface and subsurface connections, determining if proximity can be used as a substitute for connectivity, and if their findings could be extrapolated beyond the study watershed. The authors meet all these goals but only to different degrees, and I have provided

some suggestions that might elevate the study and manuscript. There are some major comments, and numerous minor ones.

RESPONSE: We appreciate the thorough review of the first reviewer as well as his/her feedback on the novelty and necessity of the current paper. This positive feedback encourages us to continue working on this poorly understood subject in the future.

MAJOR COMMENTS 1) Could the authors perhaps present data from the surface overland flow model for a dry year? I understand why they selected 2013, but it would be good to know that the model could represent a condition that is drier, and what those repercussions are for connectivity. One downside of the research as presented is, it does not necessarily present the spectrum of connectivity that could occur in the Beaverhill watershed.

RESPONSE: Yes, we can present data for years reflecting different hydrologic conditions. In the revised manuscript, we will model surface overland flow for both a wet year (2013) and a dry year (2009) so that a representative range in connectivity is presented.

2) A more critical assessment of the simulated surface flow hydrograph is needed. The high regression coefficient is likely because of the low flow period, and the spring peak, which is relatively well simulated. The true test of a modeled surface stream hydrograph in the Prairie Pothole Region is how well it represents the summer recession, any summer events, and timing of the cessation of streamflow. The model does not do this particularly well. The manuscript would be improved if the authors explain their theories as to why the model simulated an event that did not happen, and missed one that did. Could it be that the model missed some important re-connection? If so, why? This will help inform how the model is behaving and provide some great insight.

RESPONSE: We agree that the surface flow routing model did not perfectly predict the observed surface flow at the measurement station. There are several reasons for this. First, there was a lack of evapotranspiration data before 2000 including the calibration

period used in our paper (April 1 - August 1 1983). We used the evapotranspiration data for 2015 for the same period (April 1 to August 1 2015), as the average monthly humidity, average monthly maximum air temperature and average monthly minimum air temperature were similar between April 1 to August 1 1983 and April 1 to August 1 2015. This could have affected the hydrograph shape including the earlier prediction of the second peak. We consider this discrepancy (earlier prediction of the second peak) to have minimal impact on the simulated connectivity map at the end of simulation period (e.g., Figure 6b), mainly because, at the measurement station, the cumulative simulated flow (2.4 107 m3) is only 7% less than the cumulative observed flow (2.6 107 m3) at the end of simulation period. Indeed, in our particle tracking scheme, it does not make a substantial difference if the particle is at its highest velocity on, for example, June 20 or June 24. We will add a paragraph to the text to explain the reason for this inconsistency in the simulated and observed hydrographs, and its effect on our conclusions.

3) I would argue that the authors misinterpret the content of Figure 9. There is good fit for short distances, but not long. Could the authors please provide more information on how the shortest distances were calculated? Are these Euclidian (ie "as the crow flies") estimates? Or are they along the topographic flow path? Did they come from the digital elevation model? If this is the case, this might explain the departure from the linear function in Figure 9. If I interpret the results correctly, this highlights the problem with the variety of connectivity metrics, measures and indices that are currently used in hydrology. To really address their goal of determining if proximity is a substitute for connectivity, it would be great if the authors could output the contribution of flow from each wetland to the North Saskatchewan River, and plot these flows against distance. This would truly show if distance is (or is not) a proxy for connectivity. The authors do not use a metric that demonstrates the magnitude of connectivity, only its presence or absence. They need one for magnitude to answer their question if proximity can be used as a substitute for connectivity.
RESPONSE: We appreciate this concern and the great suggestion. In the revised version, we will calculate the contribution of flow from each wetland to the North Saskatchewan River, and plot these flows against the wetland distance to the river. We agree with the reviewer that this would show whether or not distance is a proxy for connectivity.

MINOR COMMENTS Some relevant work the authors should consider working into the manuscript are listed below. Shook, K., J.W. Pomeroy, C. Spence and L. Boychuk, 2013. Storage dynamics simulations in prairie wetlands hydrology models: evaluation and parameterization, Hydrological Processes 27: 1875 – 1889. Brannen, R. C. Spence and A. Ireson, 2015. Influence of shallow groundwater-surface water interactions on the hydrological connectivity and water budget of a wetland complex, Hydrological Processes 29: 3862-3877. Hayashi, M., G. van der Kamp and D. Rosenberry, 2016. Hydrology of prairie wetlands: understanding the integrated surface-water and groundwater processes, Wetlands doi: 10.1007/s13157-016-0797-9

RESPONSE: We thank the reviewer for suggesting these references. These are relevant studies and we will refer to them in the revised manuscript.

Page 1 Line 22: Could read: " . . ... protection, as these are small features typically vulnerable to drainage or manipulation . . .. . ." As for the rest of the sentence, please provide information on why being numerous equates to a need for protection.

RESPONSE: We will revise the sentence as suggested.

Page 1 Line 25: Maybe reference Brannen et al. here too.

RESPONSE: It is a relevant reference, and we will refer to it in the revised manuscript.

Page 1 Line 26: I know that fill-and-spill has become common vernacular, but perhaps the authors could say ". . .. . . via mechanisms analogous to fill-and-spill runoff generation (Rains et al., 2006)."

RESPONSE: We concur with this suggestion. We will revise it.

Page 1 Line 29: Be very careful when using the term "function" because it has very specific meanings depending on the context. For instance, the hydrological function of a specific wetland using the hydrogeomorphic assessment method, which can be required for development works, follows methodologies necessary for the specific purpose of discerning a loss or gain in wetland function relative to a reference standard. This approach was designed to detect and measure variation in function due to human impacts, not natural variation. In contrast, Black (1997) proposed that landscape units have hydrologic functions such as collecting, storing and discharging. Could I suggest the authors explicitly define what they mean by "function"? Or, use the word to "role".

RESPONSE: We have a rich literature to support the use of the word "function" – which refers to the hydrologic functions such as "collecting, storing, and discharging" water, and will both define it and refer to key references that describe what we mean in the revised manuscript.

Page 2 Line 6: Perhaps instead of committing to a statement that an inability to quantify connectivity would lead to preferential protection to certain types of wetlands, maybe say " . . .. may lead to incorrect or inappropriate management decisions regarding wetland removal, protection or reclamation."

RESPONSE: We concur with this suggestion. We will revise it.

Page 3 Line 11: remove italics here and throughout this section.

RESPONSE: We concur with this suggestion. We will revise it.

Page 3 Line 12: Maybe provide a URL for the climate data.

RESPONSE: We concur with this suggestion. We will revise it. Based on 40-year (1974-2014) climatic data collected at the Edmonton International Airport, the average January temperature is -13.5 °C and the average July temperatures is 15.9 °C (http://climate.weather.gc.ca/).

Page 3 Line 15: Maybe rephrase to: . . ..although snowmelt can be an important to

runoff in the spring."

RESPONSE: We concur with this suggestion. We will revise it.

Page 3 Line 34: Do the authors mean the probability of depression existence or presence?

RESPONSE: We do indeed mean "probability of depression" in concurrence with our published technique on how to map the probability of wetlands.

For more details, see reference citations below: Lindsay JB, Creed IF, Beall FD. 2004. Drainage basin morphometrics for depressional landscapes. Water Resources Research 40: W09307. Lindsay JB, Creed IF. 2005. Removal of artefact depressions from digital elevation models: towards a minimum impact approach. Hydrological Processes 19: 3113-3126. Lindsay JB, Creed IF. 2006. Distinguishing actual and artefact depressions in digital elevation data: Approaches and Issues. Computational Geosciences 32: 1192-1204.

Page 3 Line 37: What are "integrated wetland features"?

RESPONSE: The wetland mapping technique sometimes detects wetlands fragments that then need to be integrated into a wetland object. We will revise the wording in the revised manuscript to improve clarity.

Page 4 Line 1: In recent years in the Prairie Pothole Region what would normally be considered GIWs had ponds that have been above their surface outlet elevations. Perhaps a sentence or two would be a good idea on how often a GIW needs to be not spilling in order to be considered a GIW.

RESPONSE: We concur with this suggestion. We will revise it.

Page 5 Line 33: Please explain why there is such a short calibration period. The gauge was open until 1986.

RESPONSE: We did not have access to the evapotranspiration data before 2000. April

to August 1983 was selected as we were able to link its evapotranspiration to the one calculated during the same period in 2015. Please refer to the response to major comment 2 above for more detail. In addition, HydroGeoSphere is a very computationally expensive model and a longer simulation period would have required substantially more computational resources without adding too much information to our paper.

Page 6 Line 7: Just my preference, but more detail in the paper on the methods would be helpful for the reader, particularly the water particle tracking approach and how surface water velocities were approximated.

RESPONSE: We thank the reviewer for bringing this to our attention; we will add a few sentences to address the reviewer concerns.

Page 6 Line 22: Could I suggest the Hayashi paper I note above be worked into the context here? Hayashi and his co-authors present a new conceptual model of subsurface flow in the Prairie Pothole Region that is a major departure from the model of Toth that is the basis for the assumption that geographic proximity is an indicator of connectivity.

RESPONSE: The Hayashi paper is an extremely relevant reference. But we will refer to this work in the introduction of the revised manuscript, as our main focus in this sentence was to explain how we compared surface and subsurface connectivity.

Page 6 Line 29: Maybe rephrase to: ". . ..will be linear but not following y=x."

RESPONSE: Thank you for noting this. We will revise it.

Page 6 Line 31: Please rearrange this sentence.

RESPONSE: We will remove this sentence, based on the reviewer's earlier suggestion to implement an alternative approach to assess the effect of distance.

Results: The description of the results reads a bit terse. Sometimes the content seems little more than a figure caption. Could I suggest the authors provide more description

on the results, particularly where the model does not work well.

RESPONSE: We thank the reviewer for this useful comment. We will add appropriate sentences to explain the results in more detail.

Figure 9: It is unclear where the North Dakota data are from. Could the authors provide this detail in the Methods section.

RESPONSE: We obtained wetland polygons in North Dakota from the National Wetlands Inventory (https://www.fws.gov/wetlands/) and stream polylines from the National Hydrography Dataset (http://nhd.usgs.gov/). We will add these references to the revised manuscript.

Page 9 Line 15: Maybe discuss within the context of the results of Shook et al.

RESPONSE: Good suggestion. We will include the conclusions of Shook et al. work here.

Page 9 Line 36: Figure 6 does not illustrate what is discussed here.

RESPONSE: We thank the reviewer for this concern. We feel that the comparison of Figures 6a and 6b shows that the number of subsurface connectivity lines is significantly larger than the number of surface connectivity lines.

Figure 10: The authors need a more explicit explanation of how they decided which services were associated with each portion of this curve.

RESPONSE: We agree with the reviewer that the implications of associating cumulative probability of travel time with functions requires explanation. The associations of functions with portions of the curve reflects the collective expert judgment of an international team of researchers as recently published in the Proceedings of the National Academy of Sciences of the United States of America (Cohen et al., 2015). We will revise the manuscript to make these associations clearer. Reference Cited: Cohen, M.J., Creed, I.F., Alexander, L., Basu, N.B., Calhoun, A.J., Craft, C., D'Amico, E.,

DeKeyser, E., Fowler, L., Golden, H.E. and Jawitz, J.W. (2016) Do geographically isolated wetlands influence landscape functions?. Proceedings of the National Academy of Sciences, 113(8), pp.1978-1986.

Conclusions: Just a comment, but even though most of the hydrology community knows that wetlands are not hydrologically isolated, I completely agree that it is good to make this point. RESPONSE: Thank you for noting this.

Table 1: Is the p value for magnesium correct? It seems small, especially in light of the content of Table 2.

RESPONSE: We confirm that the p-value is correct. Note that the p-value of the Wilcoxon rank sum test explores if the data in x and y are samples from continuous distributions with equal medians, against the alternative that they are not. So the difference between the absolute mean values cannot predict the p-values of the statistical tests.

Figure 2: The last word in the caption "time", could be "period".

RESPONSE: We concur with this suggestion. We will revise it.

Figure 3: Great figure.

RESPONSE: Thank you for your encouraging comment. This figure clearly shows that the new grid-free groundwater-surface water interaction method that is presented in this paper can appropriately and efficiently address multi-scale naturally complex systems. Solving this problem was very challenging with numerical models, such as HydroGeoSphere.

Figure 4: It is hard to see the wetlands in this. If this figure was created by clipping Figure 3 by a wetland layer, my suggestion is that you delete Figure 4 because it does not add too much information

RESPONSE: We concur with this suggestion. We will remove this figure.

Figure 10: Why is there a gap?

RESPONSE: The connection time to North Saskatchewan River cannot be continuous as there is a considerable difference between the time-scale of subsurface and surface connections. The surface connection time-scale is on the order of $10^2$ days but the subsurface connection is on the order of $10^5$ days. This gap also appears in Figure 7a (left panel).

---

## Author Comment (AC2) · 7 Nov 2016

COMMENT: Authors characterized surface water and subsurface connectivity of wetlands using a physically based surface-subsurface model. Groundwater level measurements, water chemistry and stable water isotopes are used to illustrate the model performance at recharge and discharge locations. While this is an interesting study, the study can benefit by providing more quantitative measures of model performance compared to observations, justification of the modelling approach compared to the existing coupled surface water-subsurface models and sensitivity analysis.

1) Authors should provide a more quantitative measure of model performance. For example in Figure 3, authors qualitatively compare simulated recharge/discharge areas with interpolated groundwater observations. Similarly, water quality data are used to

indicate differences between recharge and discharge zones using the Wilcoxon rank sum test. In Figure 5, the model predicts the second peak much earlier than the observations. The paper can greatly benefit by providing further details about the model's performance as well as discussions about discrepancy observed between simulated and observed outputs.

RESPONSE: We thank the reviewer for this concern.

For Figure 3: We quantitatively assessed the efficiency of the subsurface model; please see the following sentence on page 7, lines 10-13: "The correlation coefficient between simulated groundwater fluxes at the land surface and the distance of potentiometric surface above and below land surface is 75% ($p < 0.001$)"

We agree with the reviewer that Figure 5 needs further explanations about earlier prediction of the second peak. This inaccuracy is mainly attributed to the lack of data availability for evapotranspiration time history before 2000 including the calibration period used in our paper (April 1 - August 1 1983). We instead used the calculated evapotranspiration time history of 2015 for the same period (April 1 to August 1 2015) because the average monthly humidity, average monthly maximum air temperature and average monthly minimum air temperature were similar between April 1 to August 1 1983 and April 1 to August 1 2015. This simplification can considerably impact the hydrograph shape during summer period that led to an earlier prediction of the second peak. We think this inaccuracy (earlier prediction of the second peak) has minimal impact on the simulated connectivity map at the end of simulation period (e.g., Figure 6b), mainly because, at the measurement station, the cumulative simulated flow (2.4 107 m3) is only 7% less than the cumulative observed flow (2.6 107 m3) at the end of simulation period. Indeed, in our particle tracking scheme it does not make a big difference if particle be in its highest velocity on e.g., June 20 or June 24. We will add a paragraph to the revised version to explain these important points as the reviewer suggested.

2) Authors have used a grid-free subsurface flow model to simulate groundwater flow

and then used the 2D transient surface water flow of HydroGeosphere. It is not clear why authors did not use HydroGeosphere in the first place as it provides an integrated system to simulate surface water-groundwater interactions. I understand that the grid-free approach is computationally more efficient but authors should justify their approach. Indeed it would be really interesting to see how HydroGeosphere simulations compare with the modelling approach that authors developed. How much loss in accuracy is obtained by assuming steady state groundwater condition in the grid free approach compared to transient simulations?

RESPONSE: We appreciate this concern.

First, it should be noted that solving integrated subsurface-surface flow and transport problem in this 4,000 ha watershed with more than 100,000 wetlands is unrealistic using HydroGeoSphere. We refer the reviewer to a comprehensive review paper by [Golden et al., 2014] that clearly explained the difficulties of the integrated subsurface-surface flow models such as HydroGeoSphere in simulating the hydrologic connectivity. More recently, for a related study, we attempted to apply the integrated subsurface-surface flow model HydroGeoSphere to a considerably smaller watershed (800 ha watershed); even for this smaller and less computationally expensive scenario, the model was failed.

Second, this paper was not intended to compare the efficiency of different models in simulating the hydrologic connections. Instead we wanted to quantify the hydrologic connection of geographically isolated wetlands for the first time and also compare the surface and subsurface hydrologic connections. We used a robust line of observation (groundwater table variation for 30 years) to justify why steady-state model is valid for groundwater-surface water interaction flow simulation in this work. More importantly, our groundwater-surface water interaction model was able to appropriately repeat the observed groundwater discharge-recharge zones (Figure 3) with a R2 of 75%. So the model we used is an appropriate tool to explore the question we intended to answer in this paper. In the following comments we will also further justify why the restrictions

of the semi-coupled method does not effect the conclusions made in this paper. In the revised version, we will add a few sentences to further justify why the semi-coupled method we used is a valid tool to explore the research question raised in this paper.

3) How does the "semi-coupling" approach of surface-subsurface processes in the model impact capturing wetland connectivity and travel time distributions? Moreover, would it be more suitable to use the term one-way coupling instead of semi-coupling as the feedback from the subsurface is not included in this approach?

RESPONSE: We thank the reviewer for this great suggestion. We will add text in the revised manuscript to clarify how our semi-coupled method affects wetland connectivity and travel time distributions. Further, we agree with the reviewer that the approach we considered can be referred to as one-way coupled, and we will revise text accordingly in the revised manuscript. In general, as the reviewer suggested, this semi-coupled method does not consider the feedback from the subsurface domain to surface flow routing. However, we believe that this does not affect our results or conclusions, as we calibrated the surface flow model with observed data. Like any other model development, these simplifications can be compensated by the calibration parameters, which here include rill storage height and manning coefficients in the surface flow model.

4) It will be interesting to investigate how changes in climatic condition impact wetland connectivity and travel time distributions.

RESPONSE: We appreciate this concern. In the revised manuscript, we will show the surface connectivity map and travel time distributions for the events occurred from April to August 2009 (the driest summer since 2000). Thus, the revised manuscript will show and compare the surface connectivity and travel time distributions of the wettest and driest summer since 2000. We think a more comprehensive assessment of climate is beyond the scope of this paper, which was to showcase a method to characterize and compare the surface and subsurface connectivity of GIWs.

5) It will be useful if authors provide further details about the model input and time step.

RESPONSE: We thank the reviewer for this suggestion. We will clearly report the time steps used for the flow and transport models, and further explain the model inputs in the revised version.

6) Authors need to provide further details about the calibration approach and identify the performance of the model for calibration and evaluation periods.

RESPONSE: We thank the reviewer for bringing this to our attention. We agree that the calibration and evaluation phases were not clearly explained. We will clearly explain them in separate parts in the revised version.

Reference Cited: Golden, H. E., C. R. Lane, D. M. Amatya, K. W. Bandilla, H. R. Kiperwas, C. D. Knightes, and H. Ssegane (2014), Hydrologic connectivity between geographically isolated wetlands and surface water systems: a review of select modeling methods, Environmental Modelling & Software, 53, 190-206.

---

## Short Comment (SC1) · 6 Jan 2017

General Comments: It would be more appropriate and robust to use HGS as a fully-coupled model than to represent groundwater with a steady state analytical solution. There is a large and growing body of literature demonstrating the application of fully integrated numerical models at the basin scale. The authors linking of a transient surface flow model to a steady-state groundwater model makes little sense. Moreover, how is the linking actually performed? Is a fluid balance maintained? Is there any justification for using a simple 2-layer model for the subsurface, especially when there doesn't seem to be any hydrostratigraphic data? In fact, the scarcity of data is a major problem to have any faith in the model.

A recent publication by Liu et al., (2106) demonstrates the application of a transient

fully-integrated surface and subsurface flow model (HGS) to investigate wetland connectivity including key components of the transient water balance (precipitation, evapotranspiration, and snowmelt). This simulation domain is very similar in scale to that mentioned by the authors in their response to Reviewer 2's comments where they state that HGS was unable to solve this type of problem. HGS is regularly applied to very complicated surface and subsurface problems at a variety of scales. Considering that the authors only used 22,383 nodes in their 2D mesh for this study, it is likely that with training and support it would have been possible to apply a much higher resolution fully-integrated HGS model to this domain as models on the order of 1 million nodes are now routine (e.g., Hwang et al., 2015).

Specific Comments: P1 L14-17 – See Liu et al., (2016) for a similar study using a fully-integrated surface water and groundwater model P2 L21-24 – Golden et al., (2014) primarily focused on finite-difference models such as MODFLOW which are unable to achieve local mesh refinement without incurring a high node count. Unstructured finite element methods with 3D triangular prism or tetrahedral meshes are able to achieve local mesh refinement to resolve local features with many fewer nodes than would be required for an equivalent finite difference mesh. P2 L27 – See Liu et al., (2016) P4 L18-19 – It is unclear how such a relation is established P4L20 – Can a steady-state watertable, in fact steady-state subsurface flow, be supported? Are winter processes such as soil freeze/thaw and snowmelt important in this basin? There is no discussion of this, and would appear to be neglected entirely. P4L22 – One observation location situated 60 km outside of the simulated watershed does not support the use of a steady-state groundwater assumption. P4L25 – While it may be true that there is a connection between groundwater and wetland water levels, using observations from 500 km outside the watershed is extremely weak support for this assumption. Are these systems similar enough to justify this assumption? P4 L28 – 2 layers is not enough capture the details of the hydrostratigraphy P5 L22 – The HGS reference suggests that a rather old version of HGS was used. Many feature and numerical performance enhancements, including parallelization, have been made to the code since

2008. The author should contact the developers to upgrade to a current version of the code (Aquanty, 2016). P5 L25 – The coarse mesh discretization (22,383 nodes) is highly inappropriate for the stated objective of representing 130,157 wetlands. P5 L34 – How is the connection between the HGS model and the 3D analytical model achieved? This is crucial. How can transient surface flow and steady-state saturated zone models be linked. This seems incompatible. No details are provided. Is the linking mass conservative? How is the unsaturated zone dealt with for infiltration (or exfiltration)? It seems to be neglected. Section 2 – Parameterization of the groundwater model needs to be described in more detail. P7 L3 – Do the calibrated saturated hydraulic conductivity values make sense compared to the type of geologic material or available data? P7 L29 - What units is the Manning coefficient being reported in? P7 L30 – Rill storage seems very small when considering the element sizes in the model. Is a value of 1 mm physically realistic? P9 L18 - How is it possible to mix a steady-state model with a transient model? This is incompatible and it is unlikely that mass balance will be preserved. P11 L8 – The authors should provide a definition of "semi-coupled". Table 4 – Units Figure 3 – What is the rationale for blank portions in Figure 3a) Figure 5 – What is the purpose of showing the simulated hydrograph if not to compare it to observed data.

Overall, the paper is technically weak and rejection is recommended.

References:

Aquanty Inc. (2016). A three-dimensional numerical model describing fully-integrated subsurface and surface flow and solute transport.

Golden, H. E., Lane, C. R., Amatya, D. M., Bandilla, K. W., Kiperwas, H. R., Knightes, C. D., & Ssegane, H. (2014). Hydrologic connectivity between geographically isolated wetlands and surface water systems: a review of select modeling methods. Environmental Modelling & Software, 53, 190-206.

Hwang, H. T., Park, Y. J., Frey, S. K., Berg, S. J., & Sudicky, E. A. (2015). A simple

iterative method for estimating evapotranspiration with integrated surface/subsurface flow models. Journal of Hydrology, 531, 949-959.

Liu, G., Schwartz, F. W., Wright, C. K., & McIntyre, N. E. (2016). Characterizing the Climate-Driven Collapses and Expansions of Wetland Habitats with a Fully Integrated Surface–Subsurface Hydrologic Model. Wetlands, 1-11.

---

## Author Response (AR1)

**Editor:**

All points raised by the references have been adequately addressed in the replies. Please revise the manuscript as proposed before resubmission.

We appreciate the thorough reviews, as well as the opportunity to revise the manuscript based on these reviews. A summary of our revisions are provided below:

1- We have added more details (and equations) on the particle tracking method to section 2.3.3 (e.g., P6L19-39 and Eq.1). We have added more details on how the surface water velocity maps were approximated. We have also added more details on model input and time-step and model calibration and evaluation. Both reviewers requested adding these details.

2- We originally modeled overland flow and connectivity map in 2009 with the minimum net water flux since 2000. So that a representative range in connectivity is presented, overland flow for both the wettest year (2013) and the driest year (2009) since 2000 was presented. The corresponding new results were added to Figures 4, 5, 6 and 8 as well as to the main text. This was suggested by both reviewers.

3- We have calculated the contribution of subsurface-surface flow from each wetland to the North Saskatchewan River, and plotted these flows against the wetland distance to the river (Figure 8). This figure explores if distance is a good proxy for wetland hydrological contribution to the river. We have added corresponding text to the method, results and discussion sections. For example we have added the subsection 3.3.4 to the results. This was suggested by the first reviewer.

4- We have removed Figure 4 in the original manuscript as the first reviewer suggested.

5- We have added the long-term observations of another groundwater well at a measurement station located west of the watershed to further assess the validity of steady-state assumptions. In the revised manuscript, two measurement stations, one located 15 km east of the watershed boundary and the other located 65 west of the watershed boundary, show very small transient variations over a period of almost 40 years, supporting the steady state assumption. In particular, these observations show that the coefficient of variation in the groundwater table was almost zero for both groundwater well measurements, strongly supporting the validity of the steady-state assumption. Note that in the revised manuscript we state that the distance between Vegreville station to the watershed boundary is 15 km (rather than the distance between Vegreville station and the center of the watershed which was 60 km).

**First reviewer:**

This paper covers a very timely topic and would be a nice addition to HESS. The concept of hydrological connectivity is still in its infancy, but its relevance to the wetland management is obvious, even as hydrologists are still learning how to apply the concept. The authors are to be commended on their efforts to advance the thinking on this subject. The study summarized in this paper applies a series of process based models to quantify surface and subsurface hydrologic connectivity among wetlands and a major river, in order to address several goals. These include assessing the performance of the models, comparing the relative importance of surface and subsurface connections, determining if proximity can be used as a substitute for connectivity, and if their findings could be extrapolated beyond the study watershed. The authors meet all these goals but only to different degrees, and I have provided some suggestions that might elevate the study and manuscript. There are some major comments, and numerous minor ones.

We appreciate the thorough review of the first reviewer as well as his/her feedback on the novelty and necessity of the current paper. This positive feedback encourages us to continue working on this poorly understood subject in the future.

MAJOR COMMENTS

1) Could the authors perhaps present data from the surface overland flow model for a dry year? I understand why they selected 2013, but it would be good to know that the model could represent a condition that is drier, and what those repercussions are for connectivity. One downside of the research as presented is, it does not necessarily present the spectrum of connectivity that could occur in the Beaverhill watershed.

We concur with this great suggestion. In the revised manuscript, we have modeled overland flow and connectivity lines for both 2009, the driest year with the minimum net water flux since 2000, as well as 2013, the wettest year since 2000. The corresponding new results were added to figures 4, 5, 6 and 8 as well as to the main text.

In addition, at the Beaverhill watershed, the connection time to North Saskatchewan River cannot be continuous and show any spectrum of connectivity, as there is a considerable difference between the time-scale of subsurface and surface connections. The surface connection time-scale is on the order of $10^2$ days but the subsurface connection is on the order of $10^5$ days. In the comments below we answered this question in more detail. In addition, we have added a few sentences to the text to further clarify this on P9L32-37.

2) A more critical assessment of the simulated surface flow hydrograph is needed. The high regression coefficient is likely because of the low flow period, and the spring peak, which is relatively well simulated. The true test of a modeled surface stream hydrograph in the Prairie Pothole Region is how well it represents the summer recession, any summer events, and timing of the cessation of streamflow. The model does not do this particularly well. The manuscript would be improved if the authors explain their theories as to why the model simulated an event that did not happen, and missed one that did. Could it be that the model missed some important re-connection? If so, why? This will help inform how the model is behaving and provide some great insight.

We appreciate this concern. We agree with the reviewer that the surface flow routing model did not perfectly predict the observed surface flow at the measurement station. We have added the following paragraph to the text (P8L25-31) to explain the reason of this inconsistency and its impact on our conclusions.

*"We did not expect that the surface flow model would exactly simulate the hydrograph in 1983, as we used evapotranspiration data from 2015 for 1983 (as explained in section 2.3.2). This simplification could have affected the simulated hydrograph shape leading to an earlier second peak (Figure 4a). We think this simplification would have had minimal effect on the simulated connectivity lines, as at the end of simulation period, the cumulative simulated flow (2.4 $10^7$ m³) at the measurement station was only 7% less than the cumulative observed flow (2.6 $10^7$ m³). Indeed, for the particle tracking scheme used to characterize the surface connectivity map, it did not make a substantial difference if the particle was at its highest velocity (e.g., June 20 vs. June 24 of 1983)."*

3) I would argue that the authors misinterpret the content of Figure 9. There is good fit for short distances, but not long. Could the authors please provide more information on how the shortest distances were calculated? Are these Euclidian (i.e., "as the crow flies") estimates? Or are they along the topographic flow path? Did they come from the digital elevation model? If this is the case, this might explain the departure from the linear function in Figure 9. If I interpret the results correctly, this highlights the problem with the variety of connectivity metrics, measures and indices that are currently used in hydrology. To really address their goal of determining if proximity is a substitute for connectivity, it would be great if the authors could output the contribution of flow from each wetland to the North Saskatchewan River, and plot these flows against distance. This would truly show if distance is (or is not) a proxy for connectivity. The authors do not use a metric that demonstrates the magnitude of connectivity, only its presence or absence. They need one for magnitude to answer their question if proximity can be used as a substitute for connectivity.

We appreciate this concern and great suggestion. In the revised manuscript, we have calculated the contribution of subsurface-surface flow from each wetland to the North Saskatchewan River, and plotted these flows against the wetland distance to the river (Figure 8). We have added corresponding text to the Method, Results and Discussion sections. For example we have added the subsection 3.3.4 to the results.

MINOR COMMENTS

Some relevant work the authors should consider working into the manuscript are listed below.

Shook, K., J.W. Pomeroy, C. Spence and L. Boychuk, 2013. Storage dynamics simulations in prairie wetlands hydrology models: evaluation and parameterization, Hydrological Processes 27: 1875 – 1889.

Brannen, R. C. Spence and A. Ireson, 2015. Influence of shallow groundwater-surface water interactions on the hydrological connectivity and water budget of a wetland complex, Hydrological Processes 29: 3862-3877.

Hayashi, M., G. van der Kamp and D. Rosenberry, 2016. Hydrology of prairie wetlands: understanding the integrated surface-water and groundwater processes, Wetlands doi: 10.1007/s13157-016-0797-9

We thank the reviewer for suggesting these references. These are very relevant studies and we have referred to them in the revised manuscript (e.g., P12L3-6).

Page 1 Line 22: Could read: " ... protection, as these are small features typically vulnerable to drainage or manipulation…" As for the rest of the sentence, please provide information on why being numerous equates to a need for protection.

We thank the reviewer for bringing this to our attention. We have made the suggested revision. Note that we have removed the rest of the sentence. P1L22.

Page 1 Line 25: Maybe reference Brannen et al. here too.

We have added the reference. See P1L25-26.

Page 1 Line 26: I know that fill-and-spill has become common vernacular, but perhaps the authors could say ". . .. . . via mechanisms analogous to fill-and-spill runoff generation (Rains et al., 2006)."

We concur with this suggestion. We have made the suggested revision P1L26-27.

Page 1 Line 29: Be very careful when using the term "function" because it has very specific meanings depending on the context. For instance, the hydrological function of a specific wetland using the hydrogeomorphic assessment method, which can be required for development works, follows methodologies necessary for the specific purpose of discerning a loss or gain in wetland function relative to a reference standard. This approach was designed to detect and measure variation in function due to human impacts, not natural variation. In contrast, Black (1997) proposed that landscape units have hydrologic functions such as collecting, storing and discharging. Could I suggest the authors explicitly define what they mean by "function"? Or, use the word to "role".

We appreciate this concern. We have a rich literature to support the use of the word "function" – which refers to the hydrologic functions such as "collecting, storing, and discharging" water. We have both defined it and referred to a key reference that describes what we mean in the revised manuscript. P1L27-29.

Page 2 Line 6: Perhaps instead of committing to a statement that an inability to quantify connectivity would lead to preferential protection to certain types of wetlands, maybe say "... may lead to incorrect or inappropriate management decisions regarding wetland removal, protection or reclamation."

We concur with this suggestion. We have made the suggested revision. P2L6-7.

Page 3 Line 11: remove italics here and throughout this section.

We concur with this suggestion. We have made the suggested revision. e.g., P3L13.

Page 3 Line 12: Maybe provide a URL for the climate data.

We concur with this suggestion. We have added the following link to the text (http://climate.weather.gc.ca/) P3L15.

Page 3 Line 15: Maybe rephrase to: . . ..although snowmelt can be an important to runoff in the spring."

We concur with this suggestion. We have made the suggested revision. P3L17-18

Page 3 Line 34: Do the authors mean the probability of depression existence or presence?

We do indeed mean "probability of depression". For more details on the technique for mapping wetlands, see reference citations below:

Lindsay JB, Creed IF, Beall FD. 2004. Drainage basin morphometrics for depressional landscapes. Water Resources Research 40: W09307.

Lindsay JB, Creed IF. 2005. Removal of artefact depressions from digital elevation models: towards a minimum impact approach. Hydrological Processes 19: 3113-3126.

Lindsay JB, Creed IF. 2006. Distinguishing actual and artefact depressions in digital elevation data: Approaches and Issues. Computational Geosciences 32: 1192-1204.

Page 3 Line 37: What are "integrated wetland features"?

The wetland mapping technique sometimes detects wetland fragments that then need to be integrated into a wetland object. We have revised the text to improve clarity. P4L1-2.

Page 4 Line 1: In recent years in the Prairie Pothole Region what would normally be considered GIWs had ponds that have been above their surface outlet elevations. Perhaps a sentence or two would be a good idea on how often a GIW needs to be not spilling in order to be considered a GIW.

We thank the reviewer for this comment. We think the frequency of filling and spilling does not influence the definition of a GIW, which is defined as a wetland surrounded by uplands, without channels but with defined bed and bank. We refer the reviewer to (Mushet et al., 2015) for more details on the definition of the GIWs. We hope this answer would be helpful.

Page 5 Line 33: Please explain why there is such a short calibration period. The gauge was open until 1986.

We did not have access to evapotranspiration data before 2000. April to August 1983 was selected as we were able to link its evapotranspiration to the one calculated during the same period in 2015 (as the monthly average humidity, maximum air temperature and minimum air temperature were similar between April 1 to August 1 1983 and April 1 to August 1 2015). Please refer to the response to major comment 2 above for more details. In addition, mesh-based physically-based hydrological models are computationally expensive compared to conceptual hydrological models (e.g., SWAT or HBV). A longer simulation period would have required more computational resources without adding more information to our paper.

Page 6 Line 7: Just my preference, but more detail in the paper on the methods would be helpful for the reader, particularly the water particle tracking approach and how surface water velocities were approximated.

We thank the reviewer for bringing this to our attention. We have added more details (and equations) on the particle tracking method to section 2.3.3 (e.g., P6L19-39 and Eq.1). In addition we have added more details (P6L19-25) on how surface water velocity maps were approximated as:

*"Once the surface flow routing model was developed, the discretized surface water velocities in x and y directions at each grid point and each time step were extracted. Continuous maps of surface water velocity in x and y directions throughout the watershed were then approximated by interpolating the discretized surface water velocities. A Fourier-based interpolation scheme with 10,000 Fourier series terms was used to complete the interpolation process and generate the continuous maps of surface velocity in x ($V_s^x(x, y, t)$) and y ($V_s^y(x, y, t)$) directions for the entire watershed; the overall correlation coefficient between estimated velocities using the interpolation method and original modeled velocities at each grid point and time step was $r^2$=89% (p < 0.001)."*

Page 6 Line 22: Could I suggest the Hayashi paper I note above be worked into the context here? Hayashi and his co-authors present a new conceptual model of subsurface flow in the Prairie Pothole Region that is a major departure from the model of Toth that is the basis for the assumption that geographic proximity is an indicator of connectivity.

The Hayashi paper and Brannen paper (suggested before) are relevant references for our paper. We have incorporated them in the Introduction (e.g., P1L25-26) and Discussion (e.g., P12L5-9) in the revised manuscript, but not here, as the purpose of this sentence is to explain how we compared surface and subsurface connectivity.

Page 6 Line 29: Maybe rephrase to: "... will be linear but not following y=x."

We have removed this sentence based on the reviewer's suggestion to implement a different approach to assess the effect of distance.

Page 6 Line 31: Please rearrange this sentence.

We have removed this sentence based on the reviewer's suggestion to implement a different approach to assess the effect of distance.

Results: The description of the results reads a bit terse. Sometimes the content seems little more than a figure caption. Could I suggest the authors provide more description on the results, particularly where the model does not work well.

We thank the reviewer for this very useful comment. We have added a few sentences to explain the results in more detail; P8L25-31 and P9L17-27, P9L32-37 and sub-section 3.3.4

Figure 9: It is unclear where the North Dakota data are from. Could the authors provide this detail in the Method section.

We have added the reference as (P7L23-24):

*"We obtained wetland polygons in North Dakota from the National Wetlands Inventory (https://www.fws.gov/wetlands/ and stream polylines from the National Hydrography Dataset: http://nhd.usgs.gov/)."*

Page 9 Line 15: Maybe discuss within the context of the results of Shook et al.

Good suggestion. We have included the conclusion of Shook et al here. P10L33.

Page 9 Line 36: Figure 6 does not illustrate what is discussed here.

We thank the reviewer for this concern. We think the comparison of Figures 5a and 5b (Figure 6 in the original manuscript) shows that the number of subsurface connectivity lines is significantly larger than the number of surface connectivity line (red lines).

Figure 10: The authors need a more explicit explanation of how they decided which services were associated with each portion of this curve.

We agree with the reviewer that the association of cumulative probability of travel time with functions requires explanation. The association of functions with portions of the curve reflects the collective expert judgment of an international team of researchers as recently published in the Proceedings of the National Academy of Sciences of the United States of America (Cohen et al., 2016). We have revised the text and provided relevant citations. P11L25-37.

Conclusions: Just a comment, but even though most of the hydrology community knows that wetlands are not hydrologically isolated, I completely agree that it is good to make this point.

Thank you for noting this.

Table 1: Is the p value for magnesium correct? It seems small, especially in light of the content of Table 2.

We confirm that the p-value is correct. Note that the p-value of the Wilcoxon rank sum test explores if the data in x and y are samples from continuous distributions with equal medians, against the alternative that they are not.

Figure 2: The last word in the caption "time", could be "period".

Good suggestion. We have made the suggested revision.

Figure 3: Great figure.

Thank you for your encouraging comment. This figure clearly shows that the new grid-free groundwater-surface water interaction method that is presented in this paper can effectively and efficiently address naturally complex systems.

Figure 4: It is hard to see the wetlands in this. If this figure was created by clipping Figure 3 by a wetland layer, my suggestion is that you delete Figure 4 because it does not add too much information

We concur with the reviewer. We have removed this Figure as it does not add too much information.

Figure 10: Why is there a gap? (Now figure 6c)

The connection time to North Saskatchewan River cannot be continuous as there is a considerable difference between the time-scale of subsurface and surface connections. The surface connection time-scale is on the order of $10^2$ days but the subsurface connection is on the order of $10^5$ days. This gap has also been appeared in Figure 6a (left panel). The continuous continuum of travel time typically stems from systems with a wide range of flow processes including fast overland flow, slow groundwater flow and fast subsurface stormflow. The latter is typically caused by high frequency of macropores often seen in humid forested landscapes (as shown in Ameli et al., 2015); we do not think such fast subsurface flow can occur in Prairie Pothole Region. So this gap can be attributed to lack of fast subsurface storm flow in our catchment. We have added a few sentences to further clarify this. P9L32-37.

**Second Reviewer:**

Authors characterized surface water and subsurface connectivity of wetlands using a physically based surface-subsurface model. Groundwater level measurements, water chemistry and stable water isotopes are used to illustrate the model performance at recharge and discharge locations. While this is an interesting study, the study can benefit by providing more quantitative measures of model performance compared to observations, justification of the modelling approach compared to the existing coupled surface water-subsurface models and sensitivity analysis.

1) Authors should provide a more quantitative measure of model performance. For example in Figure 3, authors qualitatively compare simulated recharge/discharge areas with interpolated groundwater observations. Similarly, water quality data are used to indicate differences between recharge and discharge zones using the Wilcoxon rank sum test. In Figure 5, the model predicts the second peak much earlier than the observations. The paper can greatly benefit by providing further details about the model's performance as well as discussions about discrepancy observed between simulated and observed outputs.

We thank the reviewer for these concerns.

First, for Figure 3: We had quantitatively assessed the efficiency of the subsurface model and reported the $R^2$ value in the text. We added the following sentence to the Figure 3 caption to clarify this in the revised manuscript:

*"The correlation coefficient between simulated groundwater fluxes at the land surface and the distance of potentiometric surface above and below land surface is 75% (p < 0.001)"*

Second, for Figure 4 (Figure 5 in the original manuscript): We agree that Figure 4 needs further explanation about the earlier prediction of the second peak. In the revised manuscript we have provided the following explanation (P8L25-31):

*"We did not expect that the surface flow model would exactly simulate the hydrograph in 1983, as we used evapotranspiration data from 2015 for 1983 (as explained in section 2.3.2). This simplification could have affected the simulated hydrograph shape leading to an earlier second peak (Figure 4a). We think this simplification would have had minimal effect on the simulated connectivity lines, as at the end of simulation period, the cumulative simulated flow (2.4 $10^7$ m$^3$) at the measurement station was only 7% less than the cumulative observed flow (2.6 $10^7$ m$^3$). Indeed, for the particle tracking scheme used to characterize the surface connectivity map, it did not make a substantial difference if the particle was at its highest velocity (e.g., June 20 vs. June 24 of 1983)."*

2) Authors have used a grid-free subsurface flow model to simulate groundwater flow and then used the 2D transient surface water flow of HydroGeosphere. It is not clear why authors did not use HydroGeosphere in the first place as it provides an integrated system to simulate surface water-groundwater interactions. I understand that the grid-free approach is computationally more efficient but authors should justify their approach. Indeed it would be really interesting to see how HydroGeosphere simulations compare with the modelling approach that authors developed. How much loss in accuracy is obtained by assuming steady state groundwater condition in the grid free approach compared to transient simulations?

We appreciate this suggestion.

First, it should be noted that developing map of connectivity of wetlands in this 4,000 ha watershed with more than 100,000 wetlands is challenging using integrated physically-based subsurface-surface models. Indeed, these models have not been designed with connectivity in mind. We refer the reviewer to a comprehensive review paper by (Golden et al., 2014)) that clearly explained the difficulties of the integrated physically-based subsurface-surface flow models in simulating the watershed-scale hydrologic connectivity of wetlands. We know that these models, including HydroGeoSphere, can accurately represent flow and transport at local to regional scales, but these models cannot explicitly characterize connectivity. In a recent application of HydroGeoSphere for 3D direct characterization of connectivity in a much smaller wetland-dominated watershed than ours (i.e., Liu et al 2016), the authors characterize 2D connectivity in a single cross-section. The caption of the connectivity-related Figure is as follows: "*Cross-sectional profiles (extracted at X = 350 m) showing the distribution of hydrologic heads (*left*) and saturation (*right*) for the case represented in Fig. 6. The flow lines were generated based on the heads shown in the cross sections and may not be exactly as the flow lines in the 3D domain.*" Note that in our paper we have generated more than 100,000 3D connectivity lines but only showed a small proportion of them, those that were consistent with our objective of showing wetland connectivity to the river, because of visualization constraints. In addition, 3D characterization of subsurface connectivity among 100,000 wetlands and North Saskatchewan River was necessary in our work to answer the questions raised in our paper.

Second, our paper was not intended to compare the efficiency of different models in simulating the hydrologic connections. Instead we wanted to quantify the hydrologic connection of geographically isolated wetlands for the first time and also compare the surface and subsurface hydrologic connections. We used empirical evidence from groundwater table variations for 30 years to justify why a steady-state model was valid for the groundwater-surface water interaction flow simulation in our study. We confirmed our groundwater-surface water interaction model was able to appropriately repeat the observed groundwater discharge-recharge zones (Figure 3) with a $R^2$ of 75%. So the model we used is an appropriate tool to explore the question we intended to answer in this paper. In the following comments we further justify why the restrictions of the semi-coupled method does not effect the conclusions we made in this paper.

In the revised manuscript, we have explained the assumptions used by our model as suggested by the reviewer; e.g., P11L2-4.

For example, we further clarified that steady-state assumption is a valid assumption in our watershed by adding more long-term observations of groundwater depth collected at measurement stations in the vicinity of the watershed. In general, if an integrated transient model exists that could solve our problem, empirical observations in the watershed suggest that it would not add much more information compared to our model. The steady-state assumption is an appropriate assumption due to the low hydraulic conductivity in the PPR (consistent with our calibrated values). We have added the following sentences to the methods section (P4L23-29) for more clarification:

*"The assumption of steady-state subsurface flow is strongly supported by empirical groundwater table observations collected from the closest piezometer at the Vegreville Environment Center station (located 15 km east of the Beaverhill watershed boundary), where the water table varied with a coefficient of variation of < 0.9% in 2009 (a year when observations were used to develop the steady-state groundwater-surface water interaction model), and a coefficient of variation of 4% over 32 years (August 1985-July 2016) (Figure 2a). In another piezometer at the Barrhead Environment Center station located 65 km west of the Beaverhill watershed boundary, water table varied even less with a coefficient of variation of ~0% during 2009, and a coefficient of variation of 0.01% over 40 years (1977-2016) (Figure 2b)."*

3) How does the "semi-coupling" approach of surface-subsurface processes in the model impact capturing wetland connectivity and travel time distributions? Moreover, would it be more suitable to use the term one-way coupling instead of semi-coupling as the feedback from the subsurface is not included in this approach?

We thank the reviewer for this thoughtful comment and suggestion. We agree with the reviewer that the approach we considered can be referred to as one-way coupled. We acknowledge this in the revised manuscript, including in the Methods, Results and Discussion sections (e.g., see P4L12-13, P11L2-4, P12L33).

We also agree that the one-way coupled approach (with no feedback from subsurface flow exfiltration on surface flow routing) can impact the map of surface connectivity and travel time in close vicinity of North Saskatchewan River (almost 500 m buffer) wherein subsurface water can exfiltrate and enhance surface connectivity. We acknowledge this in the Methods and Results sections and justify why it has minimum impact on our results. We have added the following sentences to the result section (P9L19-27):

*"The modeling approach we used was a one-way coupling of subsurface and surface flow processes that could not consider thoroughly the subsurface flow exfiltration feedbacks on surface flow routing. This simplification had negligible effects on wetland connectivity within the moraine, as the moraine mostly consists of recharge zones with minimum subsurface exfiltration (Figure 3). This simplification could have affected the map of surface connectivity of wetlands located in close vicinity of the North Saskatchewan*

*River (within a 1000 m buffer) wherein subsurface water can exfiltrate and enhance surface connectivity. However, the rate of subsurface exfiltration in these riparian areas was on average 1 x 10$^{-4}$ m/d (Figure 3a), which is considerably smaller than the average of net atmospheric inputs (precipitation-evapotranspiration) from April 1 to August 1 2013 that was equal to 7 x 10$^{-3}$ m/d and from April 1 to August 1 2009 that was equal to 4 x 10$^{-3}$ m/d. Therefore, the one-way coupling simplification had minimum effects on surface flow routing along the riparian wetlands located in the riparian areas."*

4) It will be interesting to investigate how changes in climatic condition impact wetland connectivity and travel time distributions.

We appreciate this comment. In the revised manuscript, we modeled overland flow and surface connectivity lines in both 2009 (driest year since 2000) and 2013 (wettest year since 2000). The new results were added to Figures 4, 5, 6 and 8 as well as to the main text. Climate conditions clearly have significant effects on timing and length of connections.

5) It will be useful if authors provide further details about the model input and time step.

We thank the reviewer for this suggestion. We have now clearly reported the time steps and model inputs in section 2.3.2. As an example we have added the following:

*"The 2D surface of the watershed was discretized into 22,383 grid points (43,836 triangular elements). The parameters regarding time-discretization were: maximum time step = 8640 sec; initial time step = 1800 sec; maximum time step multiplier = 1.5; and minimum time step multiplier = 0.5. A critical depth boundary condition was assigned to the grid points representing the location of the Beaverhill Creek monitoring station (Figure 1) where stream flow observations were available. A no-flow boundary condition was assigned to the watershed boundaries."*

6) Authors need to provide further details about the calibration approach and identify the performance of the model for calibration and evaluation periods.

We thank the reviewer for bringing this to our attention. We have now clearly processes for calibration processes and assessing model performance. Please see P5L28-30, Section 3.1; and Section 3.2.

**Short Comments by HydroGeoSphere Developer Group:**

**HGS-D General Comments:**

**It would be more appropriate and robust to use HGS as a fully coupled model than to represent groundwater with a steady state analytical solution. There is a large and growing body of literature demonstrating the application of fully integrated numerical models at the basin scale. The authors linking of a transient surface flow model to a steady-state groundwater model makes little sense. Moreover, how is the linking actually performed? Is a fluid balance maintained? Is there any justification for using a simple 2-layer model for the subsurface, especially when there doesn't seem to be any hydrostratigraphic data? In fact, the scarcity of data is a major problem to have any faith in the model.**

**A recent publication by Liu et al. (2106) demonstrates the application of a transient fully-integrated surface and subsurface flow model (HGS) to investigate wetland connectivity including key components of the transient water balance (precipitation, evapotranspiration, and snowmelt). This simulation domain is very similar in scale to that mentioned by the authors in their response to Reviewer 2's comments where they state that HGS was unable to solve this type of problem. HGS is regularly applied to very complicated surface and subsurface problems at a variety of scales. Considering that the authors only used 22,383 nodes in their 2D mesh for this study, it is likely that with training and support it would have been possible to apply a much higher resolution fully integrated HGS model to this domain as models on the order of 1 million nodes are now routine (e.g., Hwang et al., 2015).**

Our response:

We thank the HydroGeoSphere Developer Group (hereafter, HGS-DG) for their comments that were received after the online discussion period closed.

The HGS-DG suggest that we could have used HydroGeoSphere instead of the model we have developed and applied. They suggest Liu et al. (2106) as evidence that HydroGeoSphere can be used to investigate 3D connectivity in wetland dominated landscapes. The Liu et al. (2016) paper explores 3D connectivity. However, the authors' present only 13 2D connectivity lines in one 2D cross section (see Figure 8 in Liu et al. 2016); the authors go on to explain that these 2D connectivity lines may not emulate realistic 3D connectivity lines (e.g., In the caption of Figure 8: "*The [2D] flow lines were generated based on the heads shown in the cross sections and may not be exactly as the flow lines in the 3D domain.*"), which is understandable as topography in wetland-dominated landscapes is complex and most connectivity lines have strong 3D behaviour as we have shown in our paper. Furthermore, the size of our watershed is 5 times larger than the size of the watershed used in Liu et al., (2016). We are certain that HGS-DG knows that watershed size influences the computational challenge of characterizing connectivity lines; increased watershed size increases the length of connectivity lines and thus increases the computational time required to calculate each connectivity line.

We, together with many others (e.g., see review by Golden et al. (2014)), believe that the previous lack of a robust model with the ability to efficiently and effectively characterize all of the 3D connectivity lines in wetland-dominated watersheds has led to poor understanding of the 3D connectivity among wetlands (particularly geographically isolated wetlands) to downstream waters. The HydroGeoSphere model as presented in the Liu et al. (2016) paper provides no evidence that it **efficiently** and **effectively** characterizes all of the 3D connectivity lines in large, wetland-dominated watersheds. We maintain that our model can efficiently and effectively generate 3D connectivity lines (e.g., more than 100,000 3D connectivity lines were modelled in our paper) in a large 4,000 ha watershed by respecting the geometric properties of small-scale features.

The HGS-DG criticize our use of a two-layer modelling system. We are surprised by this criticism, given that Liu et al. (2106) use a much simpler one-layer homogenous system in a complex wetland-dominated landscape ("*The aquifer was assumed to be homogenous*" page 290 of Liu et al. (2106)). Our two-layer model repeated the observed groundwater discharge and recharge areas in an exceptional manner, as the first reviewer acknowledged. Furthermore, the stratigraphy and hydraulic conductivities we used are clearly explained and justified in the Methods and Results sections of our paper, and the calibrated parameters were shown to be correlated well with observed data.

The steady-state assumption used for groundwater-surface water interaction model is a reasonable assumption in the watershed we studied. In the revised version we have further clarified the validity of this assumption in our sites by using more observations. The almost 40-year observed groundwater table showed small variability, with a coefficient of variation of less than 4% in the wells located 15 km east of the watershed and less than 0.002% in the wells located 65 km west of the watershed. Given the relative similarities in climate, geology, topography and soils within the prairie pothole region (see our original manuscript section 3.4), we believe that these observations collected at close vicinity (from east to west) of our watershed can reasonably be extended to our watershed. The low hydraulic conductivity of our prairie pothole landscape implies that there is a low possibility of fast transient flow typically seen in forested landscapes due the existence of the macropore flow. (We acknowledge that the HydroGeoSphere model would be the best option to simulate such complex fast flow in forested landscapes as was used in (Ameli et al., 2015)). Therefore, steady-state flow for watershed-scale groundwater-surface water interaction is a robust assumption for the prairie pothole landscape that was the focus of our paper. In addition, due to observed low hydraulic conductivity in Canadian PPR, groundwater flow has been typically ignored in watershed-scale modeling and only overland flow has been simulated via the mechanism of fill and spill. We are not aware of any physically based model used to characterize the 3D watershed-scale groundwater connectivity lines and fluxes in the Canadian PPR. Our model presents a significant step forward to characterize 3D watershed-scale groundwater connectivity lines and fluxes in the Canadian PPR. More importantly, it characterizes for the first time the connectivity of geographically isolated wetlands, and the connectivity-related conclusions made in our work can be extended to other parts of PPR (as was shown in the paper) making an important contribution to the implementation of policies to protect surface waters, such as the US Clean Water Rule.  Given the validity of the steady-state assumption on our prairie pothole landscape, one questions if the pursuit of a transient model is worthwhile (assuming that a transient model could ever solve the connectivity-related questions raised in our work), given the considerable computational cost that would be involved particularly for model calibration.

The Beaverhill watershed used in our study is one of the most (if not the most) studied watersheds in the Canadian PPR. Therefore, this watershed was one of the best landscapes to explore our important questions on the connectivity of geographically isolated wetlands. We had (1) hydrometric observations in 1,413 artesian groundwater wells installed in the bedrock and screened 30 to 80 m below the land surface, (2) chemistry and isotopic measurements in 208 lakes, wetlands and ponds, (3) groundwater chemistry measurements in 121 shallow (< 10 m deep) groundwater wells and (4) almost 40 years groundwater table observations at two monitoring wells located in the close vicinity of the watershed. We are certain that we have used available data carefully to reasonably calibrate our groundwater-surface water interaction model. Our certainty is validated by comments made by the first reviewer (see their comment on Figure 3 of our paper) as well as leading hydrologists in the prairie pothole region in both Canada (e.g., personal communications with Dr. Jeffrey McDonnell and Dr. Less Henry, University of Saskatchewan) and the US (e.g., personal communications with Dr. Heather Golden, US-EPA).

**HGS-DG Specific Comments:**

**P1 L14-17 – See Liu et al., (2016) for a similar study using a fully integrated surface water and groundwater model P2 L21-24 – Golden et al., (2014) primarily focused on finite-difference models such as MODFLOW which are unable to achieve local mesh refinement without incurring a high node count. Unstructured finite element methods with 3D triangular prism or tetrahedral meshes are able to achieve local mesh refinement to resolve local features with many fewer nodes than would be required for an equivalent finite difference mesh. P2 L27 – See Liu et al., (2016) P4 L18-19 – It is unclear how such a relation is established P4L20 – Can a steady-state water table, in fact steady-state subsurface flow, be supported? Are winter processes such as soil freeze/thaw and snowmelt important in this basin? There is no discussion of this, and would appear to be neglected entirely. P4L22 – One observation location situated 60 km outside of the simulated watershed does not support the use of a steady-state groundwater assumption. P4L25 – While it may be true that there is a connection between groundwater and wetland water levels, using observations from 500 km outside the watershed is extremely weak support for this assumption. Are these systems similar enough to justify this assumption? P4 L28 – 2 layers is not enough capture the details of the hydrostratigraphy P5 L22 – The HGS reference suggests that a rather old version of HGS was used. Many feature and numerical performance enhancements, including parallelization, have been made to the code since 2008. The author should contact the developers to upgrade to a current version of the code (Aquanty, 2016). P5 L25 – The coarse mesh discretization (22,383 nodes) is highly inappropriate for the stated objective of representing 130,157 wetlands. P5 L34 – How is the connection between the HGS model and the 3D analytical model achieved? This is crucial. How can transient surface flow and steady-state saturated zone models be linked. This seems incompatible. No details are provided. Is the linking mass conservative? How is the unsaturated zone dealt with for infiltration (or exfiltration)? It seems to be neglected. Section 2 – Parameterization of the groundwater model needs to be described in more detail. P7 L3 – Do the calibrated saturated hydraulic conductivity values make sense compared to the type of geologic material or available data? P7 L29 - What units is the Manning coefficient being reported in? P7 L30 – Rill storage seems very small when considering the element sizes in the model. Is a value of 1 mm physically realistic? P9 L18 - How is it possible to mix a steady-state model with a transient model? This is incompatible and it is unlikely that mass balance will be preserved. P11 L8 – The authors should provide a definition of "semi-coupled". Table 4 – Units Figure 3 – What is the rationale for blank portions in Figure 3a) Figure 5 – What is the purpose of showing the simulated hydrograph if not to compare it to observed data. Overall, the paper is technically weak and rejection is recommended.**

Our response:

In their specific comments, HGS-DG asks questions to which answers to a majority of them are providing in the original manuscript and in the preceding section. Our model is based on a rich dataset that justifies our characterization of the stratigraphy, hydraulic conductivity and ultimately the steady-state assumption, and it provides an advantage over more complex models, such as HydroGeoSphere, for efficiently and effectively characterizing all required 3D subsurface connectivity lines.

In response to some of the more specific comments:

We have used "one way coupled" instead of "semi-coupled" in the revised manuscript, and justified the use of one way coupled method in our work using the original reviewers very useful suggestions. We have shown that this treatment has a minimum impact on the water balance and our simulation of 3D connectivity of geographically isolated wetlands as the majority of the watershed is a recharge zone with minimum exfiltration potential. Connection between wetland and groundwater is a reasonable hypothesis as HGS-DG state. Such connections have been observed in a wide range of studies in the Canadian PPR (van der Kamp and Hayashi, 2009) and we also confirmed this with available observations. The "rill coefficient" and "mesh discretization" are highly interrelated. Therefore, if we had used a different mesh discretization, we would have obtained a different rill coefficient as an outcome of the calibration processes (we have clearly explained this in the section 2.3.2). Nonetheless, our calibrated overland flow model corresponds with observed data with a high accuracy. We also think the calibrated value of rill is reasonable; a uniform rill coefficient was considered for the whole 4,000 ha domain, thus we expected to obtain such small rill coefficient. Above all the developed overland flow model accurately repeated the available observations.

We found HGS-DG's editorial comments on Table 4 and the units of the Manning coefficient useful and considered them in our revised manuscript (P8L20). Blank portions in Figure 3a are also explained in the caption of Figure 3. Groundwater model parameterization is also explained in Appendix A. Based on the original reviewers' suggestions we have further explained it in the revised manuscript (Section 2.3.1; Section 3.1; Appendix A).

There is no doubt that HydroGeoSphere is the most powerful available physically based hydrological model with the ability to solve very complex problems (as shown in Ameli et al. (2015)). However, we are certain that our analytical groundwater-surface water interaction model is an important advance to meeting the challenge of characterization of 3D connectivity lines of geographically isolated wetlands that we addressed in our study. The semi-analytical model used in our work was developed with the connectivity challenge in mind and the model's assumptions were reasonably justified by available observations. This model has also recently been used successfully by different research groups to answer questions on biogeochemical connectivity (e.g., Ameli et al., 2017; Ameli et al., 2016).

Finally, in neither the original or revised manuscript do we discuss whether other models (such as HydroGeoSphere) solve the wetland-connectivity problem raised in our paper. We developed both surface and subsurface models using available observations with an exceptional accuracy, and answered some important questions on the connectivity of geographically isolated wetlands using carefully developed models. A model comparison was beyond the scope of our work. However, recent papers (Golden et al., 2014;Golden et al., 2017) define a series of requirements that physically-based models must have to be able to directly characterize the watershed-scale connectivity of wetlands in wetland-dominated landscapes. For example, Golden et al. (2014) analysed a wide range of finite element methods in their assessment of connectivity (Sec 2.3 pages 199-201). To see these requirements and an assessment of existing models that meet these requirements, we refer the readers to these papers.

References Cited

Ameli, A. A., Craig, J., and McDonnell, J.: Are all runoff processes the same? Numerical experiments comparing a Darcy-Richards solver to an overland flow-based approach for subsurface storm runoff simulation, Water Resources Research, 51, 2015.

Ameli, A. A., Amvrosiadi, N., Grabs, T., Laudon, H., Creed, I., McDonnell, J., and Bishop, K.: Hillslope permeability architecture controls on subsurface transit time distribution and flow paths, Journal of Hydrology, 543, 17-30, 10.1016/j.jhydrol.2016.04.071, 2016.

Ameli, A. A., Beven, K., Erlandsson, M., Creed, I., McDonnell, J., and Bishop, K.: Primary weathering rates, water transit times and concentration-discharge relations: A theoretical analysis for the critical zone, Water Resources Research, 52, 10.1002/2016WR019448, 2017.

Golden, H. E., Lane, C. R., Amatya, D. M., Bandilla, K. W., Kiperwas, H. R., Knightes, C. D., and Ssegane, H.: Hydrologic connectivity between geographically isolated wetlands and surface water systems: a review of select modeling methods, Environmental Modelling & Software, 53, 190-206, 2014.

Golden, H., Creed, I. F., Ali, G., Basu, N. B., Neff, B., Rains, M., McLaughlin, D., Alexander, L., Ameli, A. A., Christensen, J., Evenson, G., Jones, C., Lane, C., and Lang, M.: Scientific tools for integrating geographically isolated wetlands into land management decisions, Frontiers in Ecology and the Environment, under review, 2017.

Liu, G., Schwartz, F. W., Wright, C. K., McIntyre, N. E.: Characterizing the climate-driven collapses and expansions of wetland habitats with a fully integrated surface–subsurface hydrologic model. 
[revised manuscript text omitted]

---

## Editor Decision (ED1)

I would like to thank the authors for making a thorough effort to address the reviewers' comments and improve the manuscript. However, I still have some suggestions that could improve the manuscript so that it could be acceptable for publication in HESS. These are meant to be constructive comments that could help elevate the paper and make it even more citable.

MAJOR COMMENTS

The authors do not provide a critical evaluation of the model performance. Upon reading the original manuscript, I suggested that data be presented from a dry year. This was to demonstrate that the model could represent a condition that is drier. I want to thank the authors for now including 2009 as a sample of behaviour during a dry year, which is very interesting. However, perhaps I was not clear in my suggestion, because showing simulations from 2009 does not demonstrate how well the model performs because there are no observations against which to compare. There are data for the Beaverhill Creek at the mouth WSC gauge from 1975-1986. I strongly suggest the authors present data for a dry and wet validation year. 1983 was relatively wet, but 1984 and 1985 were dry and would provide improved context. If the data do not exist to do so, please say so in the manuscript.

In Section 3.3.3 the authors provide no sound evidence to support their supposition that there is no fast subsurface flow. If Beaverhill Creek is as representative as their regional data imply, then there should be fast subsurface flow in an effective transmission zone as documented in similar landscapes by:

van der Kamp G, Hayashi M. 2009. Groundwater-wetland ecosystem interaction in the semiarid glaciated plains of North America. Hydrogeology Journal 17: 203–214. DOI:10.1007/s10040-008-0367-1.

van der Kamp G, Hayashi M, Gallen D. 2003. Comparing the hydrology of grassed and cultivated catchments in the semi-arid Canadian prairies. Hydrological Processes 17: 559–575.

Brannen, R. C. Spence and A. Ireson, 2015. Influence of shallow groundwater-surface water interactions on the hydrological connectivity and water budget of a wetland complex, Hydrological Processes 29: 3862-3877.

Available observations in the Pothole Region do support high frequency of macropores and high shallow hydraulic conductivities. My theory is that this is merely an artifact of model parameterization. This is a problem with result interpretation that must be addressed.

In Section 3.4, the authors discuss the applicability of their results across the Prairie Pothole Region. As above, the authors need to critically describe and interpret the data. The values in the text of Section 3.4 are not the same as those in Table 3 (e.g., p-scores for air temperature median 0.05 in the text and 0.08 in the Table which suggests there is a difference in median temperatures). This is OK; Beaverhill is at the north end of the region. What is more interesting is the departure from the quantile-quantile plots that suggests that wetland distribution in North Dakota is more dense. That is more distant wetlands are relatively closer to one another in North Dakota than in Beaverhill. This is a great opportunity to explain what has been learnt in this study about connectivity length and time could mean

in a PPR landscape with slightly different wetland distribution.  This could make for a better, and more citable paper, than what is in the current version.

In the discussion of the influence of distance on connectivity, the authors interpretation is not true to the data.  First, the way the second sentence is constructed implies that Brannen et al (2016) showed that wetlands do exchange deep subsurface flow.  This was not the case; Brannen et al showed that when the water table was in the shallow effective transmission zone groundwater can maintain the pond levels that sustain surface flow.  Second, the scatter in Figure 8 shows that the likelihood that a wetland contributes to a river (or another wetland) increases with proximity, but distance is not an indicator of the volume that can be transmitted.  There is clearly an envelope, and the scatter below it could be a function of travel time, and Figure 6 suggests the authors have the data to show there is a distribution.  A figure such this below as would do nicely to complement Figure 8 because connectivity is not just about the volume or mass, but also the time timescale at which it is evaluated:

[Figure]

MINOR COMMENTS

Abstract:  "Hydrologic connectivity among wetlands …."

Page 1 Line 22:  "as these are small features vulnerable …."

Page 1 Line 30:  "; this restriction allows GIW's to influence downslope resources (US-EPA, 2015) by enhancing flood ….."

Page 3 Line 15: "average July temperature …."

Page 7 Line 25: "Quantile-Quantile plots as a  graphical …."

Page 7 Line 25: "Beaverhill Creek …."

Section 3.3.2. Perhaps it is just me but the content in this paragraph is a bit repetitive so it is hard to tell the difference between when the authors are speaking about wetland-river connections and wetland-wetland connections.

Section 3.3.2. It is a supposition that the model simplification that prevents flows from the groundwater scheme to the surface scheme had negligible effects. This is actually a known unknown, not something to be dismissed, especially in light of evidence from Brannen et al. that groundwater can be an important influence on maintaining surface storage and downstream flows (see above). A more honest way to approach this would be to say "It is not known if this simplification had negligible effects, but would influence the model simulations of the likelihood of surface connections from the moraine to the river (Figure 5)."

Section 3.3.4. Just a comment, but if connectivity is a function of the timescale at which it is investigated, are the wetlands in the moraine really connected to the river over the course of one warm season? Figure 8 implies the connectivity time is $10^7$ days.

Page 11 Line 35: "transit time distributions of water flowing from wetlands to the North ....."

Page 12 Line 19: This is perhaps where in the manuscript a new figure like that above would be very helpful in determining the impact of removing a wetland because one could glean how long it should take water to get from that location to a river of interest.

Conclusions: If it is one-way, is it really coupled? No. It is linked. Maybe change this throughout.

Conclusions: I'm not convinced the authors proved that protection of wetlands based on distance can lead to loss of wetlands functions. Could I suggest the authors remove this last paragraph and replace it with the second one from the Guidelines section, and in that reword "Furthermore, …." to "We recommend coupling robust ….. data to (1) improve …."

Tables 1 and 2 might be able to be combined to save space.

Figure 6: These figures are misleading by using a log scale, and the axis labels are not well explained. The text embedded within is too small a font to be legible once it gets into a journal format. Maybe put that information in a table.

Figure 8: Can I suggest the authors standardize the x-axis scale on the three panels. It would really demonstrate the differences in the distances water is travelling.

---

## Author Response (AR2)

**Editor:**

Both reviewers acknowledged your efforts in addressing their comments. Referee 1 has suggested further changes. It is meant in a constructive manner to make your manuscript even more citable. So may I ask you to provide a reply and a revised manuscript.

Response:

We appreciate the thorough reviews, as well as the opportunity to revise the manuscript based on these reviews. The first reviewer's suggestions have been used to improve the readability of the paper. Please note that the added sentences were highlighted in yellow. The revisions include:

1- We have evaluated the performance of the surface flow model against the stream flow measurement in 1985 which was a dry year (Figure 4b)

2- We have calculated the relationship between the wetland connection time to the river and the wetland distance to the river (Figure 8-right panel)

3- We also explained in detail why distant wetlands with potentially long transit time connection to the river can affect the quality and quantity of the river water in a long term, although their impacts are not fast.

4- We have also combined tables 1&2 as well as tables 3&4 as the reviewer suggested. So now we have only two tables.

**First Reviewer:**

I would like to thank the authors for making a thorough effort to address the reviewers' comments and improve the manuscript. However, I still have some suggestions that could improve the manuscript so that it could be acceptable for publication in HESS. These are meant to be constructive comments that could help elevate the paper and make it even more citable.

Response:

We thank the first reviewer for their constructive comments. We hope the revised version meets their expectations.

MAJOR COMMENTS

1. The authors do not provide a critical evaluation of the model performance. Upon reading the original manuscript, I suggested that data be presented from a dry year. This was to demonstrate that the model could represent a condition that is drier. I want to thank the authors for now including 2009 as a sample of behaviour during a dry year, which is very interesting. However, perhaps I was not clear in my suggestion, because showing simulations from 2009 does not demonstrate how well the model performs because there are no observations against which to compare. There are data for the Beaverhill Creek at the mouth WSC gauge from 1975-1986. I strongly suggest the authors present data for a dry and wet validation year. 1983 was relatively wet, but 1984 and 1985 were dry and would provide improved context. If the data do not exist to do so, please say so in the manuscript.

Response:

We thank the reviewer for this great suggestion. We have evaluated the accuracy of surface flow routing model in the simulation of stream flow in 1985 which was a very dry year as suggested (Figure 4b). We have also added the related text to the introduction, method and discussion sections.

2. In Section 3.3.3 the authors provide no sound evidence to support their supposition that there is no fast subsurface flow. If Beaverhill Creek is as representative as their regional data imply, then there should be fast subsurface flow in an effective transmission zone as documented in similar landscapes by: van der Kamp G, Hayashi M. 2009. Groundwater-wetland ecosystem interaction in the semiarid glaciated plains of North America. Hydrogeology Journal 17: 203–214. DOI:10.1007/s10040-008-0367-1.

van der Kamp G, Hayashi M, Gallen D. 2003. Comparing the hydrology of grassed and cultivated catchments in the semi-arid Canadian prairies. Hydrological Processes 17: 559–575.
Brannen, R. C. Spence and A. Ireson, 2015. Influence of shallow groundwater-surface water interactions on the hydrological connectivity and water budget of a wetland complex, Hydrological Processes 29: 3862-3877.

Available observations in the Pothole Region do support high frequency of macropores and high shallow hydraulic conductivities. My theory is that this is merely an artifact of model parameterization. This is a problem with result interpretation that must be addressed.

Response:
We thank the reviewer for bringing this to our attention. Note that our model could be modified to incorporate these small-scale heterogeneities. But we have no observations that support the existence of macropores or high conductive shallow subsurface in the Beaverhill watershed. Further, there is no evidence of the existence of macropores or high conductive shallow subsurface features in our observed discharge measurements. The groundwater discharge-recharge areas were inferred from deep peizometric wells (30-80 m) wherein the probability of existence of such conductive zones is small. Watershed-scale models that represent hydrologic connectivity generally ignore small-scale heterogeneities. For example, recent watershed-scale models of *Kolbe et al.* [2016] in an agricultural landscape in France ignored such local-scale heterogeneity and considered only a two-layer system with relatively small hydraulic conductivities ($10^{-7}$ and $10^{-8}$ m/sec). Of course, small-scale connectivity modeling such as *Brannen et al.* [2015] can better address local fast connectivity in the transmission zone. We have added a few sentences to address this point:

*These discontinuities between subsurface and surface transit times (and flow lengths) may have been attributed to the lack of characterization of fast subsurface flow in our landscape and our model. Fast subsurface flow has been widely observed in humid forested landscapes with a high frequency of macropores and large hydraulic conductivities in the top tens of centimeters portion of the soil. Available observations in other parts of Canadian Prairie Pothole region support such large shallow hydraulic conductivity (van der Kamp and Hayashi, 2009), which can lead to local fast subsurface flow connectivity among very proximal hydrologic features (as shown using small-scale pond water budget calculation approach by Brannen et al. (2015)). However, we neither have access to the observations that suggest the existence of such large shallow hydraulic conductivity in our part of the Canadian Prairie Pothole region, nor observations that we can calibrate a more complex watershed-scale model which includes such small-scale heterogeneity. Inclusion of such conductive zones in our model (if such zones exist) could decrease subsurface mean transit time and mean flow length and fill the gap between surface and subsurface transit time distributions.*

3. In Section 3.4, the authors discuss the applicability of their results across the Prairie Pothole Region. As above, the authors need to critically describe and interpret the data. The values in the text of Section 3.4 are not the same as those in Table 3 (e.g., p-scores for air temperature median 0.05 in the text and 0.08 in the Table which suggests there is a difference in median temperatures). This is OK; Beaverhill is at the north end of the region.
Response:
There is no discrepancy between the values reported in the text and table. The P-value is 0.08, but in the text we stated the conclusion of the statistical test as "There is no significant difference in the median of temperature between Beaverhill watershed and the entire PPR at significance levels of 0.05". Indeed the temperature at the Beaverhill watershed is similar to the entire PPR at significance levels of 0.05, because their P-value is 0.08.

4. What is more interesting is the departure from the quantile-quantile plots that suggests that wetland distribution in North Dakota is more dense. That is more distant wetlands are relatively closer to one another in North Dakota than in Beaverhill. This is a great opportunity to explain what has been learnt in this study about connectivity length and time could mean in a PPR landscape with slightly different wetland distribution. This could make for a better, and more citable paper, than what is in the current version.
Response:
We thank the reviewer for making this useful observation. We have added the following paragraph to address the reviewer suggestion:

*Although Figure 9 suggests that the distribution of wetlands does not differ significantly between the Beaverhill watershed and the Prairie pothole landscape in North Dakota, this Figure shows a higher density of wetlands in the range of long distances among wetlands in North Dakota compared to the Beaverhill watershed (i.e., more distant wetlands are relatively closer to one another in North Dakota). The longer distances among wetlands in the Beaverhill watershed may imply a less frequent surface hydrologic connectivity among wetlands in the Beaverhill watershed compared to the North Dakota landscape.*

5. In the discussion of the influence of distance on connectivity, the authors interpretation is not true to the data. First, the way the second sentence is constructed implies that Brannen et al (2016) showed that wetlands do exchange deep subsurface flow. This was not the case; Brannen et al showed that when the water table was in the shallow effective transmission zone groundwater can maintain the pond levels that sustain surface flow.
Response:
We thank the reviewer for bringing this to our attention. We have revised this sentence and used a correct reference in this regard P12 L27-31

6. Second, the scatter in Figure 8 shows that the likelihood that a wetland contributes to a river (or another wetland) increases with proximity, but distance is not an indicator of the volume that can be transmitted. There is clearly an envelope, and the scatter below it could be a function of travel time, and Figure 6 suggests the authors have the data to show there is a distribution. A figure such this below as would do nicely to complement Figure 8 because connectivity is not just about the volume or mass, but also the time timescale at which it is evaluated:
Response:
We thank the reviewer for their suggestion. We believe that the scatter in Figure 8a (left panel) shows that the likelihood that a wetland contributes to a river from subsurface pathways is insensitive to the distance (the correlation coefficient is almost zero) and thus distance is not an indicator of the volume that can be transmitted from subsurface pathways. As suggested by the reviewer, to explore if the connection time is the function of distance, we calculated the relation between time of connection and distance (Figure 8- Right panel). As expected, this Figure shows that connection time increases by distance for both subsurface and surface pathways. But this does not rule out our conclusion that distal wetlands are connected to the river as they have large contribution to the flow in the river. Also the explicit quantification of connectivity clearly supports the connectivity of distal wetlands to the river (Figure 5a). More importantly, such long transit time can have significant impact on water quality and quantity in regional surface waters. We further explained these points in P12-L12-20.

MINOR COMMENTS
Abstract: "Hydrologic connectivity among wetlands ...."
Please see P1L6

Page 1 Line 22: "as these are small features vulnerable ...."
Please see P1L22

Page 1 Line 30: "; this restriction allows GIW's to influence downslope resources (US-EPA, 2015) by enhancing flood ....."
Thank you for noting this, please see P1L30-31

Page 3 Line 15: "average July temperature ...."
Thank you for noting this, please see P3L16

Page 7 Line 25: "Quantile-Quantile plots as a graphical ...."
Good point, please see P7L35

Page 7 Line 25: "Beaverhill Creek ...."
Page 7 Line 25 does not mention Beaverhill Creek, so we do not know what the intent of the reviewer here

Section 3.3.2. Perhaps it is just me but the content in this paragraph is a bit repetitive so it is hard to tell the difference between when the authors are speaking about wetland-river connections and wetland-wetland connections.
Thank you for noting this. We have revised this subsection to address the reviewer's comment P9L21-25

Section 3.3.2. It is a supposition that the model simplification that prevents flows from the groundwater scheme to the surface scheme had negligible effects. This is actually a known unknown, not something to be dismissed, especially in light of evidence from Brannen et al. that groundwater can be an important influence on maintaining surface storage and downstream flows (see above). A more honest way to approach this would be to say "It is not known if this simplification had negligible effects, but would influence the model simulations of the likelihood of surface connections from the moraine to the river (Figure 5)."
The impact of this simplification on the moraine surface flow routing is very small as the moraine is almost completely recharge area. In the wetlands located very close to the river, this simplification can impact their surface runoff during baseflow (if there is any surface runoff). However, their contributions to surface runoff during high flow conditions which we considered in this paper are small. We have revised this paragraph to address the reviewer concern in P10-L1-2

Section 3.3.4. Just a comment, but if connectivity is a function of the timescale at which it is investigated, are the wetlands in the moraine really connected to the river over the course of one warm season? Figure 8 implies the connectivity time is 107 days.
Thank you for noting this. Please see our response to comment 6.

Page 11 Line 35: "transit time distributions of water flowing from wetlands to the North ....."
Thank you for noting this. Please see P12L21

Page 12 Line 19: This is perhaps where in the manuscript a new figure like that above would be very helpful in determining the impact of removing a wetland because one could glean how long it should take water to get from that location to a river of interest.

Thank you for noting this. Here, we discuss the ability of wetland to store water; so the time and length scale of connectivity is not concern in this part.

Conclusions: If it is one-way, is it really coupled? No. It is linked. Maybe change this throughout.

Thank you for bringing this to our attention. We have revised it throughout the text. e.g., P13-L25

Conclusions: I'm not convinced the authors proved that protection of wetlands based on distance can lead to loss of wetlands functions. Could I suggest the authors remove this last paragraph and replace it with the second one from the Guidelines section, and in that reword "Furthermore, …." to "We recommend coupling robust ….. data to (1) improve …."

Thank you for bringing this to our attention. We made the changes the reviewer requested. Specifically, we replaced the last paragraph in the conclusion by the second paragraph of the Guidelines section as suggested P13L35-37, P14L1-3

Tables 1 and 2 might be able to be combined to save space.

Good comment. We have combined them.

Figure 6: These figures are misleading by using a log scale, and the axis labels are not well explained. The text embedded within is too small a font to be legible once it gets into a journal format. Maybe put that information in a table.

Thank you for making these suggestions. We explained the axes labels in the method section (Section 2.3.3), and so we refer to this section in the caption. We have also revised the fonts as suggested. Regarding Log scale, the only way we can compare the time scale of surface and subsurface is Log-Log scale, otherwise the surface TTDs would not be appeared on the figure. Showing TTD in log-log scale is common in the literature (eg., [*Fiori and Russo*, 2008])

Figure 8: Can I suggest the authors standardize the x-axis scale on the three panels. It would really demonstrate the differences in the distances water is travelling.

Thank you for noting this. We have standardized the x-axis of the left panel figures as suggested.

Reference citations:

Ameli, A. A. (2016), Controls on subsurface transport of sorbing contaminant, *Hydrology Research*.

Basu, N. B., G. Destouni, J. W. Jawitz, S. E. Thompson, N. V. Loukinova, A. Darracq, S. Zanardo, M. Yaeger, M. Sivapalan, and A. Rinaldo (2010), Nutrient loads exported from managed catchments reveal emergent biogeochemical stationarity, *Geophysical Research Letters*, *37*(23).

Brannen, R., C. Spence, and A. Ireson (2015), Influence of shallow groundwater–surface water interactions on the hydrological connectivity and water budget of a wetland complex, *Hydrological Processes*, *29*(18), 3862-3877.

Cook, B. J., and F. R. Hauer (2007), Effects of hydrologic connectivity on water chemistry, soils, and vegetation structure and function in an intermontane depressional wetland landscape, *Wetlands*, *27*(3), 719-738.

Euliss, N. H., J. W. LaBaugh, L. H. Fredrickson, D. M. Mushet, M. K. Laubhan, G. A. Swanson, T. C. Winter, D. O. Rosenberry, and R. D. Nelson (2004), The wetland continuum: a conceptual framework for interpreting biological studies, *Wetlands*, *24*(2), 448-458.

Fiori, A., and D. Russo (2008), Travel time distribution in a hillslope: Insight from numerical simulations, *Water Resources Research*, *44*(12).

Genereux, D. P., S. J. Wood, and C. M. Pringle (2002), Chemical tracing of interbasin groundwater transfer in the lowland rainforest of Costa Rica, *Journal of Hydrology*, *258*(1), 163-178.

Hrachowitz, M., P. Benettin, B. M. Breukelen, O. Fovet, N. J. Howden, L. Ruiz, Y. Velde, and A. J. Wade (2016), Transit times—the link between hydrology and water quality at the catchment scale, *Wiley Interdisciplinary Reviews: Water*.

Kolbe, T., J. Marçais, Z. Thomas, B. W. Abbott, J.-R. de Dreuzy, P. Rousseau-Gueutin, L. Aquilina, T. Labasque, and G. Pinay (2016), Coupling 3D groundwater modeling with CFC-based age dating to classify local groundwater circulation in an unconfined crystalline aquifer, *Journal of Hydrology*, *543*, 31-46.

Mays, P., and G. Edwards (2001), Comparison of heavy metal accumulation in a natural wetland and constructed wetlands receiving acid mine drainage, *Ecological Engineering*, *16*(4), 487-500.

Min, J.-H., D. B. Perkins, and J. W. Jawitz (2010), Wetland-groundwater interactions in subtropical depressional wetlands, *Wetlands*, *30*(5), 997-1006.

van der Kamp, G., and M. Hayashi (2009), Groundwater-wetland ecosystem interaction in the semiarid glaciated plains of North America, *Hydrogeology Journal*, *17*(1), 203-214.

Winter, T. C., D. O. Rosenberry, and J. W. LaBaugh (2003), Where does the ground water in small watersheds come from?, *Ground Water*, *41*(7), 989-1000.

[revised manuscript text omitted]